# Circuit and synaptic organization of forebrain-to-midbrain pathways that promote and suppress vocalization

Valerie Michael[1], Jack Goffinet[1], John Pearson[1,2], Fan Wang[1], Katherine Tschida[3]*, Richard Mooney[1]*

[1]Department of Neurobiology, Duke University Medical Center, Durham, United States; [2]Department of Biostatistics & Bioinformatics, Duke University Medical Center, Durham, United States; [3]Department of Psychology, Cornell University, Ithaca, United States

**Abstract** Animals vocalize only in certain behavioral contexts, but the circuits and synapses through which forebrain neurons trigger or suppress vocalization remain unknown. Here, we used transsynaptic tracing to identify two populations of inhibitory neurons that lie upstream of neurons in the periaqueductal gray (PAG) that gate the production of ultrasonic vocalizations (USVs) in mice (i.e. PAG-USV neurons). Activating PAG-projecting neurons in the preoptic area of the hypothalamus ($POA_{PAG}$ neurons) elicited USV production in the absence of social cues. In contrast, activating PAG-projecting neurons in the central-medial boundary zone of the amygdala ($Amg_{C/M-PAG}$ neurons) transiently suppressed USV production without disrupting non-vocal social behavior. Optogenetics-assisted circuit mapping in brain slices revealed that $POA_{PAG}$ neurons directly inhibit PAG interneurons, which in turn inhibit PAG-USV neurons, whereas $Amg_{C/M-PAG}$ neurons directly inhibit PAG-USV neurons. These experiments identify two major forebrain inputs to the PAG that trigger and suppress vocalization, respectively, while also establishing the synaptic mechanisms through which these neurons exert opposing behavioral effects.

*For correspondence:
kat227@cornell.edu (KT);
mooney@neuro.duke.edu (RM)

**Competing interests:** The authors declare that no competing interests exist.

## Introduction

The decision to vocalize is often a matter of life and death, as vocalizations are an important medium for sexual and social signaling between conspecifics but may also inadvertently advertise the caller's location to eavesdropping predators. Consequently, many factors influence the decision to vocalize, including the presence of external sensory and social cues, as well as the animal's own internal state and past experience. Work from the last five decades has established the midbrain periaqueductal gray (PAG) as an obligatory gate for the production of vocalizations in all mammals (*Fenzl and Schuller, 2002*; *Jürgens, 1994*; *Jürgens, 2002*; *Jürgens, 2009*; *Subramanian, et al., 2020*; *Sugiyama et al., 2010*; *Tschida et al., 2019*), and it is thought that forebrain inputs to the PAG regulate the production of vocalizations in a context-dependent fashion. In line with this idea, forebrain regions including the cortex, amygdala, and hypothalamus have been implicated in regulating vocalization as a function of social context (*Bennett et al., 2019*; *Dujardin and Jürgens, 2006*; *Gao et al., 2019*; *Green et al., 2018*; *Jürgens, 1982*; *Jürgens, 2002*; *Kyuhou and Gemba, 1998*; *Ma and Kanwal, 2014*; *Manteuffel et al., 2007*). Notably, although electrical or pharmacological activation of various forebrain regions can elicit vocalizations (*Jürgens, 2009*; *Jürgens and Ploog, 1970*; *Jürgens and Richter, 1986*), these effects depend on an intact PAG (*Jürgens and Pratt, 1979*; *Lu and Jürgens, 1993*; *Siebert and Jürgens, 2003*), suggesting that the PAG acts as an essential hub for descending forebrain control of vocalization. Despite the centrality of the PAG to

vocalization, the synaptic mechanisms through which forebrain neurons interact with the PAG vocal gating circuit to either promote or suppress vocalization remain unknown.

A major challenge to understanding the synaptic mechanisms through which descending forebrain neurons influence vocalization was that, until recently, the identity of the PAG neurons that play an obligatory role in vocal gating remained unknown. The PAG is a functionally heterogeneous structure important to many survival behaviors (*Bandler and Shipley, 1994*; *Carrive, 1993*; *Evans et al., 2018*; *Holstege, 2014*; *Tovote et al., 2016*), thus hindering the identification of vocalization-related PAG neurons and forebrain inputs to these neurons that might influence vocalization. To overcome this challenge, we recently used an intersectional activity-dependent genetic tagging technique to identify neurons in the PAG of the mouse that gate the production of ultrasonic vocalizations (USVs; i.e. PAG-USV neurons *Tschida et al., 2019*), which mice produce in a variety of social contexts (*Chabout et al., 2015*; *Holy and Guo, 2005*; *Maggio and Whitney, 1985*; *Neunuebel et al., 2015*; *Nyby, 1979*; *Portfors and Perkel, 2014*; *Whitney et al., 1974*). The identification of PAG-USV neurons opens the door to identifying their monosynaptic inputs and to understanding how these afferent synapses modulate neural activity within the PAG vocal gating circuit to influence vocal behavior.

Here, we combined intersectional methods and transsynaptic tracing to identify neurons that provide monosynaptic input to PAG-USV neurons and to local PAG inhibitory interneurons. Using this transsynaptic tracing as an entry point, we then identified inhibitory neurons in both the hypothalamus and the amygdala that provide synaptic input to the PAG vocal gating circuit. In male and female mice, we found that optogenetic stimulation of hypothalamic afferents to the PAG-USV circuit promoted USV production in the absence of any social cues, whereas similar stimulation of amygdalar afferents to the PAG-USV circuit in males suppressed spontaneous USV production elicited by social encounters with females. Lastly, we used optogenetic-assisted circuit mapping in brain slices to deduce the synaptic mechanisms through which these forebrain afferents act on PAG-USV neurons and PAG interneurons to exert their opposing effects on USV production. This study provides the first functional description of the synaptic logic that governs the decision to vocalize, a behavior fundamental to communication and survival.

## Results

### Inhibitory neurons in the hypothalamus and amygdala provide input to the PAG vocal gating circuit

To identify forebrain neurons that provide input to the PAG vocal gating circuit, we performed transsynaptic tracing from PAG-USV neurons, which are primarily glutamatergic and reside in the caudolateral PAG (*Tschida et al., 2019*). Briefly, to label inputs to PAG-USV neurons, we used an activity-dependent labeling strategy (CANE; see Materials and methods) to express Cre-dependent helper viruses in PAG-USV neurons (*Rodriguez et al., 2017*; *Sakurai et al., 2016*; *Tschida et al., 2019*). A pseudotyped replication-deficient rabies virus was subsequently injected into the caudolateral PAG, allowing selective transsynaptic labeling of direct inputs to PAG-USV neurons (*Figure 1A*, see Materials and methods). Due to the difficulty in eliciting robust and reliable USVs from female mice, transsynaptic tracing from CANE-tagged PAG-USV neurons was performed only in male mice. Because the activity of many glutamatergic PAG neurons is shaped by potent inhibition from GABAergic PAG neurons (*Tovote et al., 2016*), we also performed transsynaptic tracing from local GABAergic neurons in the caudolateral and ventrolateral PAG that likely provide inhibition onto PAG-USV neurons, an idea we confirmed in a later section of the Results. To label direct inputs to local GABAergic PAG interneurons, Cre-dependent helper viruses were injected into the caudo/ventrolateral PAG of a VGAT-Cre mouse, and a pseudotyped replication-deficient rabies virus was subsequently injected at the same site to enable transsynaptic tracing from these cells (*Figure 1B*; N = 4 males, N = 2 females). These rabies tracing experiments revealed monosynaptic inputs to PAG-USV neurons as well as to local GABAergic PAG neurons from a variety of forebrain areas (*Figure 1—source data 1*, *Figure 1—figure supplements 1–4*). We subsequently focused on thoroughly characterizing forebrain afferents from the hypothalamus and amygdala, two brain regions important for the regulation and production of emotional and social behaviors, including vocalization

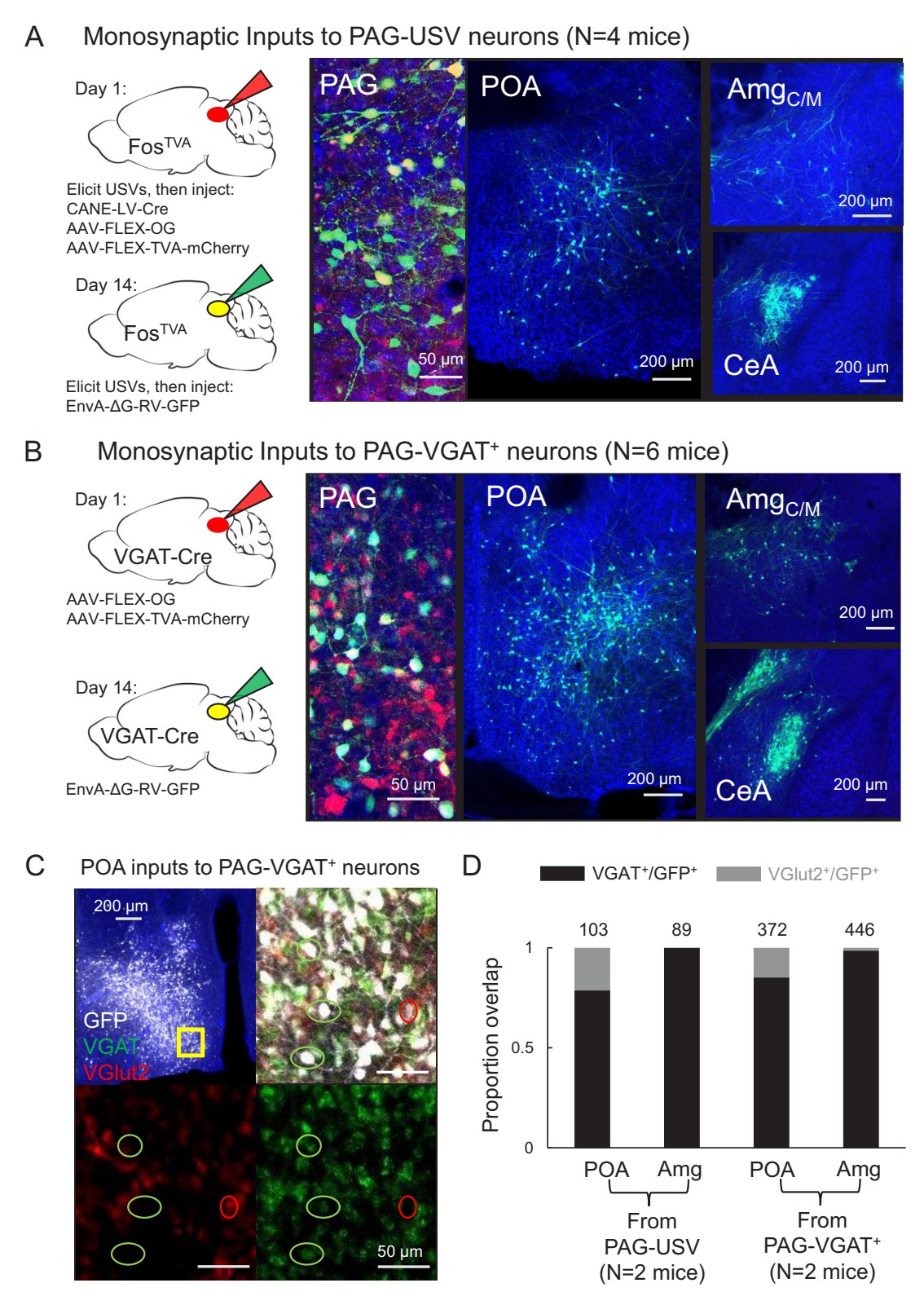

**Figure 1.** Inhibitory neurons in the hypothalamus and amygdala provide input to the periaqueductal gray (PAG) vocal gating circuit. (A) (Left) Viral strategy shown for transsynaptic labeling of direct inputs to PAG-USV neurons (performed in N = 4 males). (Right) Confocal images are shown of starter PAG-USV neurons, upstream neurons labeled within the preoptic area of the hypothalamus (POA), and upstream neurons labeled within the Amg$_{C/M}$ and CeA. (B) Same, for transsynaptic labeling of direct inputs to GABAergic PAG neurons (performed in N = 4 males, N = 2 females). (C)

*Figure 1 continued on next page*

*Figure 1 continued*

Representative confocal image of in situ hybridization performed on transsynaptically labeled POA neurons that provide direct input to GABAergic PAG neurons (labeled with GFP, shown in white), showing overlap with expression of VGAT (green) and VGlut2 (red). DAPI shown in blue. (D) Quantification of overlap of transsynaptically labeled POA and amygdala neurons (CeA and Amg$_{C/M}$ combined) with VGAT and VGlut2 (N = 2 male mice per condition). Total numbers of neurons scored for overlap in each condition are indicated by the numbers over the bars. See also *Figure 1— figure supplements 1–4*, and *Figure 1—source datas 1–2*.

The online version of this article includes the following source data and figure supplement(s) for figure 1:

**Source data 1.** Source data for *Figure 1A–C*.
**Source data 2.** Source data for *Figure 1D*.
**Figure supplement 1.** Monosynaptic rabies-based tracing reveals preoptic and amygdala inputs to the midbrain vocalization circuit.
**Figure supplement 2.** Monosynaptic rabies-based tracing reveals cortical inputs to the midbrain vocalization circuit.
**Figure supplement 3.** Monosynaptic rabies-based tracing reveals additional hypothalamic inputs to the midbrain vocalization circuit.
**Figure supplement 4.** Monosynaptic rabies-based tracing reveals additional subcortical inputs to the midbrain vocalization circuit.

(*Chen and Hong, 2018*; *Duvarci and Pare, 2014*; *Ehrlich et al., 2009*; *Gothard, 2020*; *Janak and Tye, 2015*; *LeDoux, 2007*; *Sternson, 2013*).

Within the hypothalamus, we observed labeling of neurons in the medial preoptic area (POA, *Figure 1A–B*), a region that plays a crucial role in sexual behavior (*Balthazart and Ball, 2007*; *McKinsey et al., 2018*; *Newman, 1999*; *Wei et al., 2018*) and more specifically in the production of courtship vocalizations in rodents and in songbirds (*Alger and Riters, 2006*; *Bean et al., 1981*; *Floody, 1989*; *Floody, 2009*; *Floody et al., 1998*; *Fu and Brudzynski, 1994*; *Gao et al., 2019*; *Riters and Ball, 1999*; *Vandries et al., 2019*). Within the amygdala, we observed inputs to both PAG cell types from neurons spanning the rostral portion of the boundary between the central and medial amygdala (referred to here as the central-medial boundary zone (Amg$_{C/M}$, see below) continuing caudally to the central amygdala (CeA) (*Figure 1A–B*). Although the amygdala contributes to the sensory processing of and behavioral responses to social and emotional vocalizations (*Fecteau et al., 2007*; *Gadziola et al., 2016*; *Hall et al., 2013*; *Schönfeld et al., 2020*), whether and how the amygdala contributes to the *production* of vocalizations remains understudied (see *Hall et al., 2013*; *Ma and Kanwal, 2014*; *Matsumoto et al., 2012*).

To characterize the neurotransmitter phenotypes of these upstream hypothalamic and amygdala neurons, we performed two-color in situ hybridization on transsynaptically labeled neurons for mRNA transcripts expressed in glutamatergic and GABAergic cells (vesicular glutamate transporter (vGluT2) and vesicular GABA transporter (VGAT); *Figure 1C*, see Materials and methods). This experiment revealed that the majority (~84%, *Figure 1C–D*) of PAG-projecting POA neurons (i.e. POA$_{PAG}$ neurons) and almost all (~98%, *Figure 1D*) of PAG-projecting Amg$_{C/M}$ and CeA neurons (i.e. Amg$_{C/M-PAG}$ and CeA$_{PAG}$ neurons) are GABAergic. In summary, the PAG vocal gating circuit receives input from inhibitory neurons in both the preoptic hypothalamus and the amygdala.

## Activating PAG-projecting POA neurons elicits USVs in the absence of social cues

The POA plays a crucial role in courtship, raising the possibility that POA$_{PAG}$ neurons are important to promoting USV production. To test this idea, we selectively expressed channelrhodopsin (ChR2) in POA$_{PAG}$ neurons by injecting a Cre-dependent AAV driving ChR2 expression into the POA and injecting a retrogradely infecting AAV that drives Cre expression into the caudolateral PAG, the region in which PAG-USV neurons are concentrated (*Figure 2A*). Optogenetic activation of POA$_{PAG}$ cell bodies was sufficient to elicit USVs in male and female mice that were singly tested in the absence of social partners or social cues (USVs elicited in N = 6 of 8 males, N = 3 of 4 females; 10 Hz trains or tonic pulses of 1–2 s duration; *Figure 2A–B*, *Video 1*). Although optogenetic activation of POA$_{PAG}$ neurons often elicited robust USV production, the efficacy of optogenetic stimulation (number of USVs elicited per trial, number of successful trials) as well as the latency from stimulation to USV onset were variable both within and across individual mice (*Figure 2B and F*, and *Figure 2— figure supplement 1*). This vocal effect was specifically attributable to optogenetic activation of the POA, as delivery of blue light to the POA of GFP-expressing mice failed to elicit USVs (AAV-FLEX-GFP injected into the POA of Esr1-Cre males, see below for additional Esr1-Cre data, N = 5, *Figure 2F*). In summary, optogenetic activation of POA$_{PAG}$ neurons is capable of promoting USV

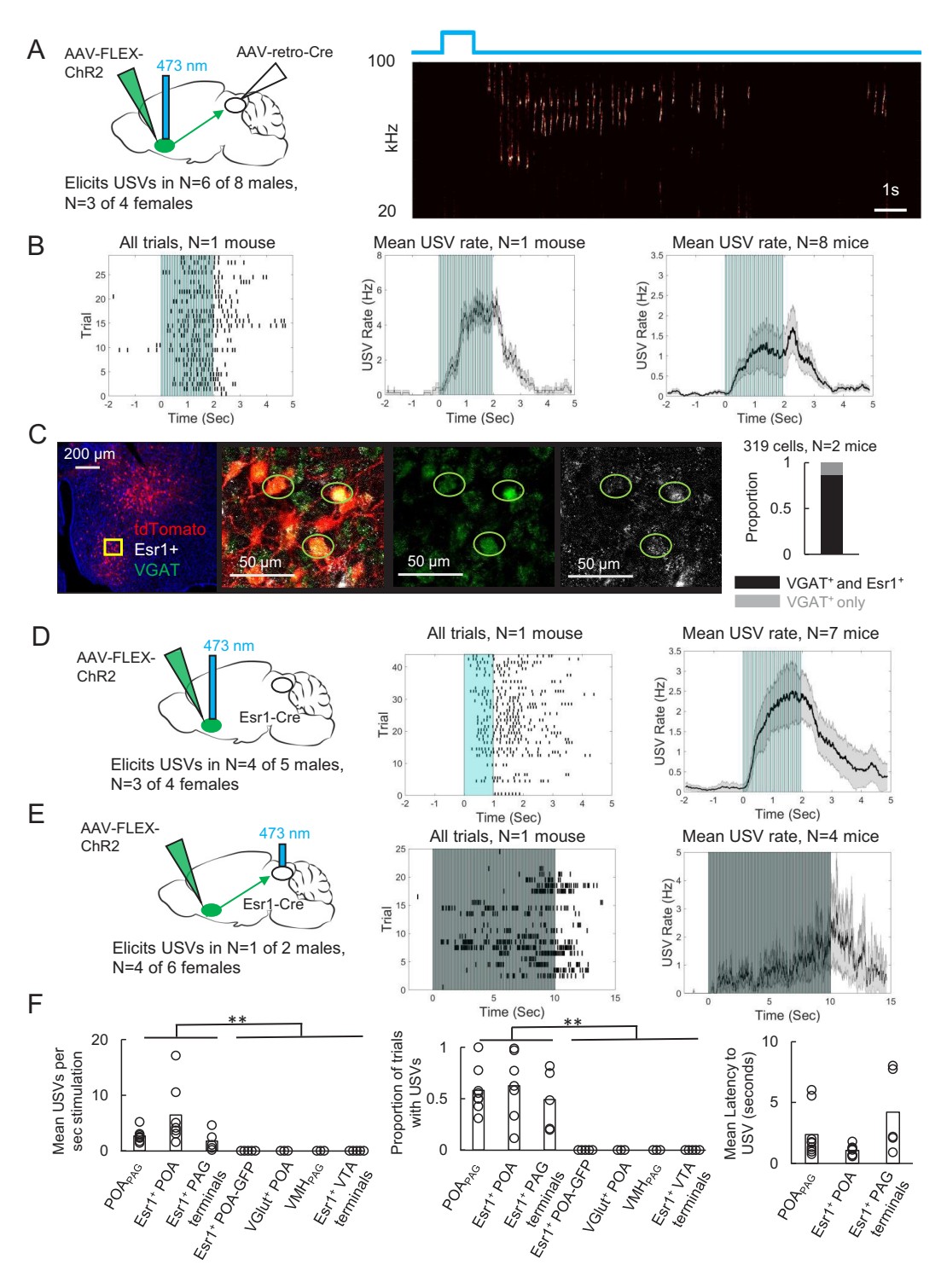

**Figure 2.** Activating periaqueductal gray (PAG)-projecting POA neurons elicits ultrasonic vocalizations (USVs) in the absence of social cues. (A) (Left) Viral strategy to express ChR2 in POA_PAG neurons. (Right) Example trial showing that optogenetic activation of POA_PAG neurons elicits USV production in an isolated animal. (B) (Left) Raster plot shows USVs elicited in many trials in a representative mouse following optogenetic activation of POA_PAG neurons. (Middle) Mean USV rate aligned to delivery of blue light pulses plotted for that same mouse. (Right) Mean USV rate plotted for N = 8 mice following optogenetic activation of POA_PAG neurons. Please note that one mouse in which USVs were elicited by optogenetic stimuli that did not include the 2s-long, 10 Hz stimulus is excluded from the summary analysis shown in the right-most panel. Gray shading above and below the mean in the middle and right panels represents S.E.M. See also *Figure 2—figure supplement 1*. (C) Representative confocal image and quantification of in situ

*Figure 2 continued on next page*

*Figure 2 continued*

hybridization performed on POA$_{PAG}$ neurons (tdTomato, red), showing overlap with Esr1 (white) and VGAT (green). DAPI is blue, N = 2 mice. (**D**) (Left) Viral strategy used to express ChR2 in Esr1$^+$ POA neurons. (Middle) Raster plot shows USVs elicited in many trials in a representative mouse following optogenetic activation of Esr1$^+$ POA neurons. (Right) Mean USV rate plotted for N = 7 mice following optogenetic activation of Esr1$^+$ POA neurons. Gray shading above and below the mean represents S.E.M. (**E**) Same as (**D**), for experiments in which the axon terminals of Esr1$^+$ POA neurons were optogenetically activated within the PAG. Data shown for stimulation with10s-long, 20 Hz blue light pulses. Please note that one mouse in which USVs were elicited by optogenetic stimuli that did not include the 10s-long, 20 Hz stimulus is excluded from the summary analysis shown in the right-most panel. (**F**) Summary plots show mean number of USVs per second of optogenetic stimulation (left, p=0.0013, one-way ANOVA between all groups, with post-hoc t-tests showing that each experimental condition was significantly different from control conditions at p<0.01), mean number of optogenetic trials with USVs (middle, p=1.8E-6, one-way ANOVA between all groups, with post-hoc t-tests showing that each experimental condition was significantly different from control conditions at p<0.01), and mean latency from onset of optogenetic stimulus to onset of first USV (right) for mice in which optogenetic stimulation was applied to POA$_{PAG}$ neurons (N = 9 mice), Esr1$^+$ POA neurons (N = 7 mice), Esr1$^+$ POA axon terminals within the PAG (N = 5 mice), GFP-expressing Esr1$^+$ POA neurons (N = 5 mice), VGlut$^+$ POA neurons (N = 3 mice), VMH$_{PAG}$ neurons (N = 3), and Esr1+ POA axon terminals within the ventral tegmental area (VTA) (N = 4 mice). See also *Figure 2—figure supplements 1–3* and *Figure 2—source data 1*.

The online version of this article includes the following source data and figure supplement(s) for figure 2:

**Source data 1.** Source data for *Figure 2C and F*.
**Figure supplement 1.** Additional characterization of ultrasonic vocalizations (USVs) elicited by optogenetic activation of preoptic area (POA) neurons.
**Figure supplement 1—source data 1.** Source data for panels B and C of *Figure 2—figure supplement 1*.
**Figure supplement 2.** Additional information related to the optogenetic activation of POA$_{PAG}$, POA-Esr1+ neurons, and Amg$_{C/M-PAG}$ neurons.
**Figure supplement 2—source data 1.** Source data for panel A of *Figure 2—figure supplement 2*.
**Figure supplement 3.** Dual tracing of the axonal projections of POA$_{PAG}$ and Amg$_{C/M-PAG}$ neurons.

production in both male and female mice, consistent with the known role of the POA in promoting appetitive courtship behaviors.

To begin to describe the molecular phenotype of POA$_{PAG}$ neurons, we used situ hybridization to establish that these cells express VGAT (319/319 neurons were VGAT$^+$; *Figure 2C*), similar to the POA neurons that we labeled via transsynaptic tracing from the PAG vocal gating circuit. We also noted that the majority of POA$_{PAG}$ neurons co-express Estrogen Receptor 1 (Esr1), a prominent marker for neurons in the POA (278/319 *Figure 2C*; *Fang et al., 2018*; *Moffitt et al., 2018*; *Wei et al., 2018*). Given that POA$_{PAG}$ neurons express Esr1, we next tested whether optogenetic activation of Esr1$^+$ POA neurons was sufficient to elicit USV production, by injecting a Cre-dependent AAV driving the expression of ChR2 into the POA of Esr1-Cre mice. We observed that optogenetically activating Esr1$^+$ POA neurons was sufficient to elicit USV production in male (N = 4 of 5) and female mice (N = 3 of 4) that were tested in the absence of any social partners or social cues (*Figure 2D*). In contrast, optogenetic activation of VGlut2$^+$ neurons within the POA failed to elicit USV production (*Figure 2F*, N = 3 males, POA of VGlut-Cre mice injected with AAV-FLEX-ChR2). Our findings confirm and extend the recent finding that optogenetic activation of GABAergic POA neurons elicits USV production in male and female mice (*Gao et al., 2019*).

To test whether activation of the Esr1$^+$ POA neurons that project to the PAG is sufficient to elicit USVs, we optogenetically activated the axon terminals of Esr1$^+$ POA neurons within the PAG (*Figure 2E*). Bilateral Esr1$^+$ POA$_{PAG}$ terminal activation within the PAG was sufficient to elicit USV production (N = 1 of 2 males; N = 4 of 6

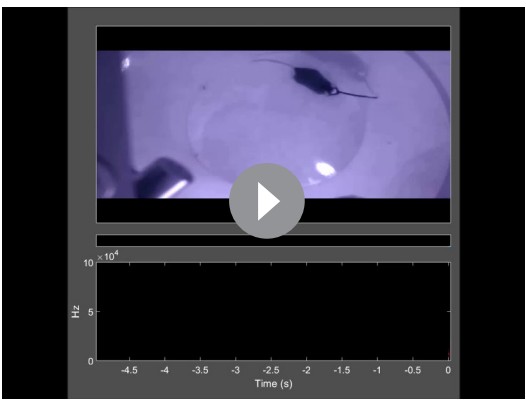

**Video 1.** Optogenetic activation of POA$_{PAG}$ neurons elicits ultrasonic vocalizations (USVs). An isolated male mouse is shown which has ChR2 is expressed in POA$_{PAG}$ neurons. Optogenetic activation of these neurons with pulses of blue light elicits USV production. Video is shown at the top, a spectrogram (bottom) showing the optogenetically elicited USVs is synchronized to the video, and pitch-shifted audio (80 kHz to 5 kHz transformation) is included to place the ultrasonic vocalizations (USVs) within the human hearing range.
https://elifesciences.org/articles/63493#video1

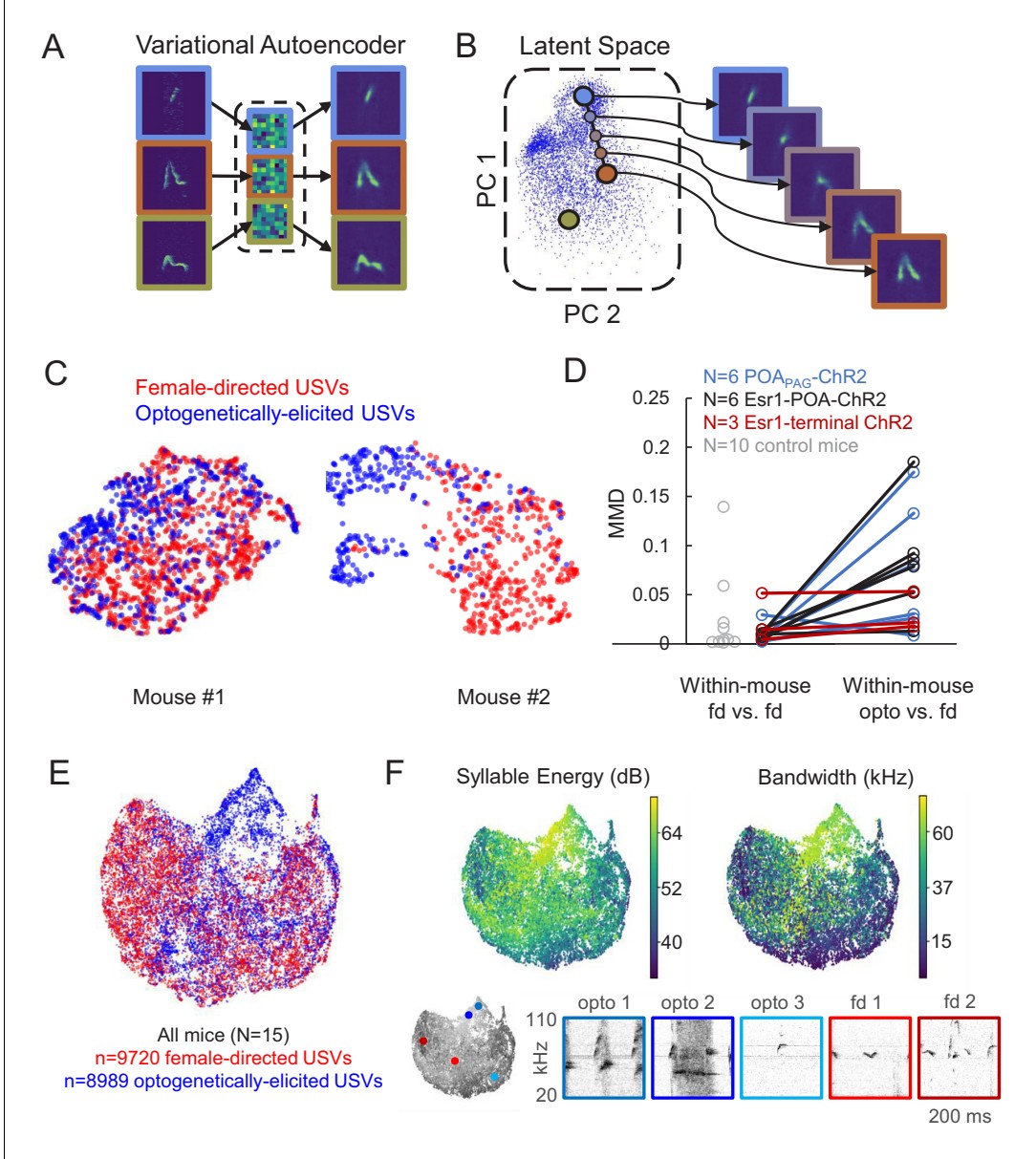

**Figure 3.** Acoustic characterization of ultrasonic vocalizations (USVs) elicited by optogenetic activation of preoptic area (POA) neurons. (**A**) The variational autoencoder (VAE) takes spectrograms as input (left), maps the spectrograms to low-dimensional latent representations using an encoder (middle), and approximately reconstructs the input spectrogram using a decoder (right). (**B**) (Left) Dimensionality reduction techniques such as PCA or UMAP can be used to visualize the resulting latent vectors. (Right) Interpolations in latent space correspond to smooth USV syllable changes in spectrogram space exhibiting realistic dimensions of variation. (**C**) UMAP projections of latent syllable representations of female-directed USVs (red) and optogenetically elicited USVs (blue) from two example mice. (**D**) Maximum Mean Discrepancy (MMD) was calculated between distributions of latent syllable representations to generate three comparisons: female-directed USVs from the first half of the recording session vs. female-directed USVs from the second half, all female-directed USVs vs. opto-USVs (N = 16 experimental mice), and female-directed USVs from two different recordings sessions in N = 10 control mice (gray points). Note that larger MMD values indicate that distributions are more dissimilar (**E**) UMAP projections of latent descriptions of female-directed (red) and optogenetically elicited USVs (blue) for all mice (N = 15). (**F**) UMAP projections from panel E, color-coded by total energy (left) and frequency bandwidth (right). Example spectrograms of opto-USVS and female-directed USVs are plotted below, and the location of each example USV in UMAP space is indicated by the colored dots on the grayscale UMAP projection on the bottom left. See also *Figure 3—source data 1*.

The online version of this article includes the following source data for figure 3:

**Source data 1.** Source data for *Figure 3D*.

females, 20 Hz trains of 2–10 s duration). This treatment also evoked escape behavior in four of eight of the tested animals, which was not observed following optogenetic activation of Esr1$^+$ POA cell bodies, suggesting that viral spread to PAG-projecting neurons nearby to the POA may account for these effects. Finally, we sought to test the idea that optogenetic activation of Esr1$^+$ POA neurons promotes USV production through their projections to the PAG rather than through other regions that they also innervate, and we also tested whether USV production could be elicited by activating non-POA hypothalamic inputs to the PAG. In fact, USVs were not elicited by optogenetically activating either Esr1$^+$ POA axon terminals in the ventral tegmental area (VTA) (*Figure 2F*, 0/2 females, 0/2 males) or PAG-projecting neurons within the ventromedial hypothalamus (VMH) (*Figure 2F*, 0/3 males, AAV-retro-Cre injected in the PAG, AAV-FLEX-ChR2 injected in the VMH). Therefore, GABAergic POA neurons, including Esr1$^+$ cells, act via their synapses in the caudolateral PAG to promote USV production.

Although these control experiments are consistent with the idea that POA$_{PAG}$ and Esr1$^+$ POA neurons act directly on the vocal gating mechanism in the PAG, a remaining possibility is that they promote USV production through hedonic reinforcement. To control for this possibility, we performed real-time place preference tests in which optogenetic stimulation of either POA$_{PAG}$ or Esr1$^+$ POA neurons was applied when mice were in only one of two sides of the test chamber. We observed that optogenetic activation of POA$_{PAG}$ neurons drove a slightly negative place preference on average (mean PP = 0.39 +/- 0.07 for N = 7 mice; *Figure 2—figure supplement 2*, panel A) and that optogenetic activation of Esr1$^+$ POA cell bodies did not positively reinforce place preference (mean PP = 0.46 +/- 0.02 for N = 5 mice). In contrast, optogenetic activation of Esr1$^+$ POA axon terminals within the VTA positively reinforced place preference (mean PP = 0.59 +/- 0.06, N = 4 mice, *Figure 2—figure supplement 2*, panel A). We also note that when using the same stimulation parameters that were sufficient to elicit USVs, optogenetic activation of either POA$_{PAG}$ or Esr1$^+$ POA neurons did not drive mounting of other mice (N = 7 POA$_{PAG}$-ChR2 mice tested; N = 4 POA-Esr1-ChR2 mice tested) nor did it induce overt locomotion (*Figure 2—figure supplement 2*, panel B). These experiments indicate that activation of POA$_{PAG}$ neurons can elicit USVs in a manner that does not depend on positive reinforcement and without recruiting other courtship behaviors.

Because the POA lies upstream of the PAG, we anticipated that optogenetic activation of the POA would elicit USV production at longer latencies than observed for optogenetic activation of PAG-USV neurons. Indeed, we found that the minimum and mean latencies to elicit USVs by optogenetic stimulation of POA neurons were 664.5 +/- 320.9 ms and 1782.6 +/- 407.6 ms, respectively (*Figure 2—figure supplement 1*, calculated from N = 9 POA$_{PAG}$-ChR2 and N = 7 POA-Esr1-ChR2 mice). These latencies are longer than those observed when optogenetically activating PAG-USV neurons (PAG-USV activation: min. latency from laser onset to first USV was 23.4 ± 8.6 ms, mean latency was 406.6 ± 0.5 ms) (*Tschida et al., 2019*) but are comparable to the latencies from optogenetic activation of the hypothalamus to observed effects on behavior that have been reported in other studies (*Lin et al., 2011*; *Wei et al., 2018*). We also found that USV bouts elicited by optogenetic activation of the POA often outlasted the duration of the optogenetic stimulation, sometimes by many seconds (*Figure 2*, *Figure 2—figure supplement 1*). This contrasts with what is observed following optogenetic activation of PAG-USV neurons, in which USV bout durations map on tightly to the duration of optogenetic stimulation (*Tschida et al., 2019*), and suggests that brief optogenetic stimulation in the POA can be transformed into longer lasting changes in neural activity within the POA or across POA-to-PAG synapses.

## Acoustic characterization of USVs elicited by activation of POA neurons

Given that optogenetic stimulation of the POA elicited USVs in the absence of any social cues, we wondered whether such optogenetically evoked USVs were acoustically similar to the USVs that mice produce during social interactions. To compare the acoustic features of optogenetically elicited USVs to those of USVs produced spontaneously to a nearby female, we employed a recently described method using variational autoencoders (VAEs) (*Goffinet et al., 2019*; *Sainburg et al., 2019*). Briefly, the VAE is an unsupervised modeling approach that uses spectrograms of vocalizations as inputs and from these data learns a pair of probabilistic maps, an 'encoder' and a 'decoder,' capable of compressing vocalizations into a small number of latent features while attempting to preserve as much information as possible (*Figure 3A–B*). Notably, this method does not rely on user-defined acoustic features, nor does it require clustering of vocalizations into categories. We applied

this approach to spectrograms of USVs to compare the acoustic features of female-directed and optogenetically elicited USVs from the same mice and found that the VAE converged on a concise latent representation of only five dimensions. We then employed a dimensionality reduction method (UMAP) (*McInnes et al., 2018*) to visualize the latent features of these USVs in 2D space (*Figure 3C*). This analysis revealed that for some mice, female-directed and optogenetically elicited USVs were acoustically similar (*Figure 3C*, left), while for other mice, a subset of optogenetically elicited USVs were acoustically distinct from female-directed USVs (*Figure 3C*, right). To quantify the difference between female-directed and optogenetically elicited USVs for each mouse, we estimated the Maximum Mean Discrepancy (MMD) (*Gretton et al., 2012*) between distributions of latent syllable representations as in *Goffinet et al., 2019*. In addition, a baseline level of variability in syllable repertoire was established for each mouse by estimating the MMD between the first and second halves of female-directed USVs emitted in a recording session (*Figure 3D*). A paired comparison revealed significantly larger differences between female-directed and optogenetically elicited USVs than expected by variability within the female-directed recording sessions alone (*Figure 3D*, two-sided, continuity-corrected Wilcoxon signed-rank test, W = 5, p<0.01), or than expected by across-day variability in female-directed recording sessions from control animals (gray points, *Figure 3D*, female-directed USVs were recorded on 2 different days from N = 10 control mice, p=0.003 for difference between female-directed vs. female-directed in control mice and opto vs. female-directed in experimental mice, Mann Whitney U test). In conclusion, many USVs elicited by optogenetic activation of POA neurons resemble female-directed USVs, although a subset differs in their acoustic features from USVs found in the animals' female-directed repertoires.

We next sought to understand in more detail exactly how these acoustically unusual optogenetically elicited USVs differed from natural USVs. When the latent representations of these two types of USVs were plotted together for all mice in our dataset, it became clear that optogenetically-elicited USVs and female-directed USVs are largely acoustically overlapping except in one region of the UMAP representation (upper middle portion of *Figure 3E*, dominated by blue points). Despite this outlying region of acoustically distinct optogenetically elicited USVs, we conservatively estimate that only 20% of condition information (optogenetically elicited versus female-directed) can be predicted by latent syllable descriptions, consistent with largely overlapping distributions of natural and optogenetically elicited USVs (0.20 bits, fivefold class-balanced logistic regression). We then re-plotted UMAP representations of the USVs, with each USV syllable color-coded according to syllable energy (i.e. amplitude, *Figure 3F*, left) or frequency bandwidth (*Figure 3F*, right). This analysis revealed that the acoustically unusual optogenetically elicited USVs tended to be louder and had greater frequency bandwidths than female-directed USVs. Visual inspection of spectrograms of optogenetically-elicited USVs also confirmed that those that did not overlap acoustically with natural USVs tended to be louder and have greater frequency bandwidths (*Figure 3F*, bottom, opto 1 and opto 2), while optogenetically elicited USVs that overlapped with natural USVs did not possess these unusual acoustic features (*Figure 3F*, bottom, opto 3). To determine whether the differences between optogenetically elicited and natural USVs were consistent across mice, we summarized each recording session by the mean latent representation of its syllables, and then summarized the shift from natural to optogenetically elicited syllable repertoires by the corresponding vector between summary points. A shuffle test revealed significantly larger alignment between these vectors than expected by chance (mean cosine similarity = 0.50, p<1e-5), indicating that optogenetically-elicited USVs differed from female-directed USVs in a manner that was consistent across mice. In summary, optogenetic activation of the POA elicits USVs whose acoustic features are largely overlapping with those of female-directed USVs produced by the same animal, despite the artificiality inherent to optogenetic stimulation.

## Activating PAG-projecting Amg$_{C/M}$ neurons transiently suppresses USV production

We then explored how PAG-projecting amygdala neurons contribute to vocalization. We began with a viral strategy designed to express ChR2 in PAG-projecting Amg$_{C/M}$ and CeA neurons, by injecting a Cre-dependent AAV driving ChR2 expression targeted to the amygdala and then injecting AAV-retro-Cre into the PAG (*Figure 4*). Surprisingly, given the strong transsynaptic labeling of both the CeA and Amg$_{C/M}$ achieved with modified rabies tracing from the PAG vocal gating circuit, we found that this viral strategy failed to label neurons in the CeA and instead only labeled Amg$_{C/M}$ neurons,

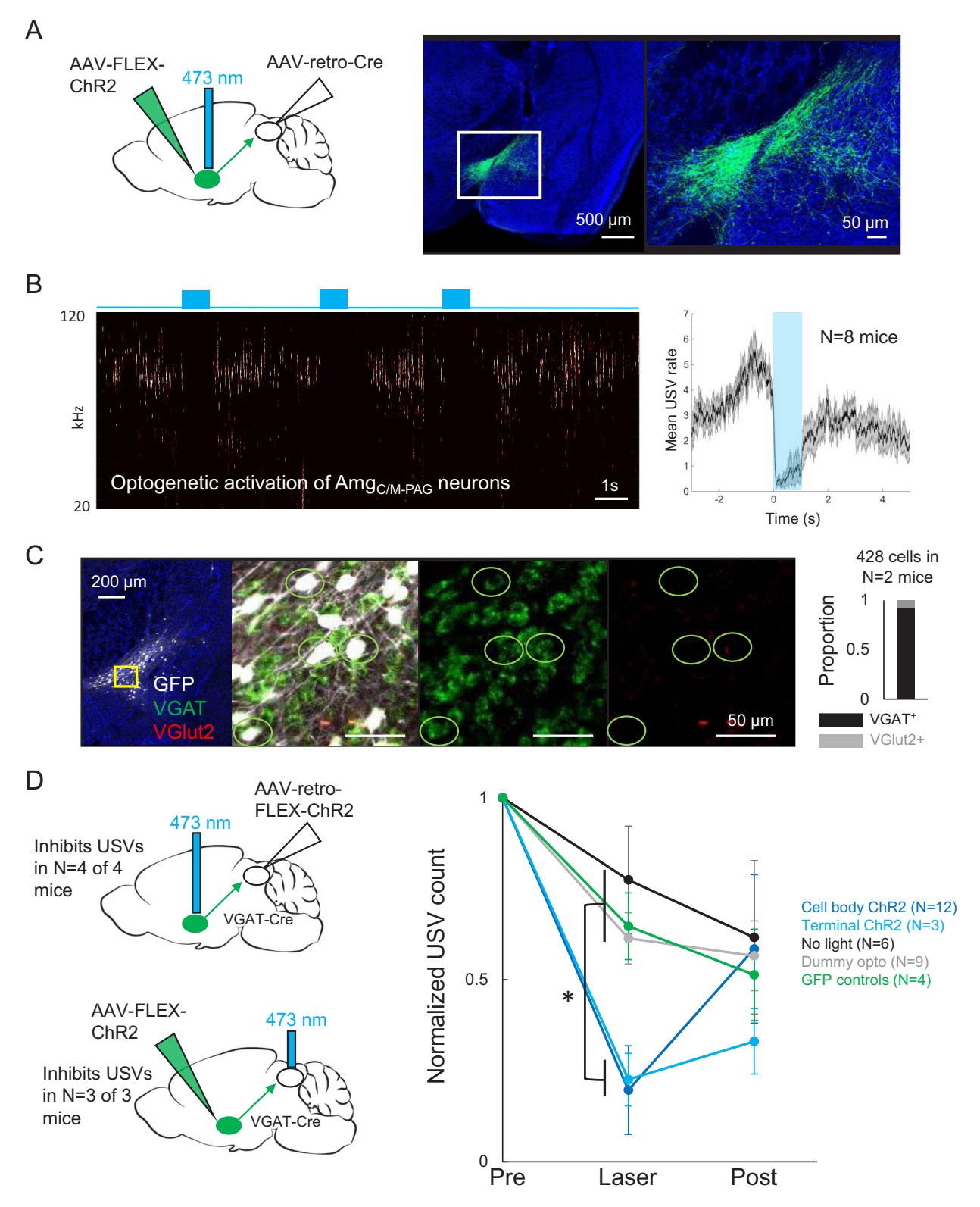

**Figure 4.** Activating Amg$_{C/M-PAG}$ neurons transiently suppresses ultrasonic vocalization (USV) production. (**A**) (Left) Viral strategy used to express ChR2 in Amg$_{C/M-PAG}$ neurons. (Right) Confocal image of representative Amg$_{C/M-PAG}$ cell body labeling achieved with this viral strategy. (**B**) (Left) Spectrogram showing a representative trial in which optogenetic activation of Amg$_{C/M-PAG}$ neurons suppresses USV production during the laser stimulation period. (Right) Group data quantified for N = 8 mice. Gray shading above and below the mean represents S.E.M. (**C**) Confocal image and quantification of in

*Figure 4 continued on next page*

*Figure 4 continued*

situ hybridization performed on Amg$_{C/M-PAG}$ neurons (GFP, shown in white), showing overlap with VGlut2 (red) and VGAT (green). DAPI in blue, N = 2 mice. (**D**) Left: viral strategy used to express ChR2 in the periaqueductal gray (PAG) axon terminals of Amg$_{C/M-PAG}$ neurons. Right: Quantification of the number of USVs produced in the 1 s period prior to optogenetic stimulation (pre), the 1 s period of optogenetic stimulation (laser), and the 1 s period following optogenetic stimulation (post). Data for each mouse were normalized by dividing the pre, laser, and post measurements by the total number of USVs produced during the pre-laser period. Group averages are shown for mice in which Amg$_{C/M-PAG}$ neurons were optogenetically activated (N = 12, dark blue), mice in which the PAG axon terminals of Amg$_{C/M-PAG}$ neurons were optogenetically activated (N = 3, light blue), control mice in which the blue laser was shined over the mouse's head but not connected to the optogenetic ferrule (N = 9, gray), control mice in which GFP was expressed in Amg$_{C/M-PAG}$ neurons (N = 4, green), and control mice in which the laser was triggered but not turned on (N = 6, black). Error bars represent S.D. Please note that the decay in USV rates over time in the control groups reflects the natural statistics of USV production (increasing probability that a bout will end as time progresses). See also *Figure 4—figure supplements 1–2*, *Figure 2—figure supplements 1–2*, and *Figure 4—source data 1*.

The online version of this article includes the following source data and figure supplement(s) for figure 4:

**Source data 1.** Source data for *Figure 4C and D*.
**Figure supplement 1.** Extent of cell body labeling of Amg$_{C/M-PAG}$ neurons.
**Figure supplement 2.** Comparison of hypothalamus and amygdala cell body labeling achieved after transsynaptic tracing from the periaqueductal gray (PAG) vocal gating circuit versus.

whose cell bodies reside medial to the CeA and dorsal to the medial amygdala (*Figure 4A*, *Figure 4—figure supplement 1*). To ensure that this labeling pattern was due to restricted tropism of the AAV-retro-Cre virus and not to inaccurate targeting of the CeA, we repeated the injections of the AAV-retro-Cre virus in the PAG of a Cre-dependent tdTomato reporter mouse. Again, we observed cell body labeling in the Amg$_{C/M}$ but not in the CeA (*Figure 4—figure supplement 2*), suggesting that in contrast to the modified rabies virus used in the transsynaptic tracing from the PAG vocal gating circuit, the AAV-retro-Cre virus can infect Amg$_{C/M}$ but not CeA neurons.

To test whether PAG-projecting Amg$_{C/M}$ neurons influence USV production, we first tested the effects of optogenetically activating these

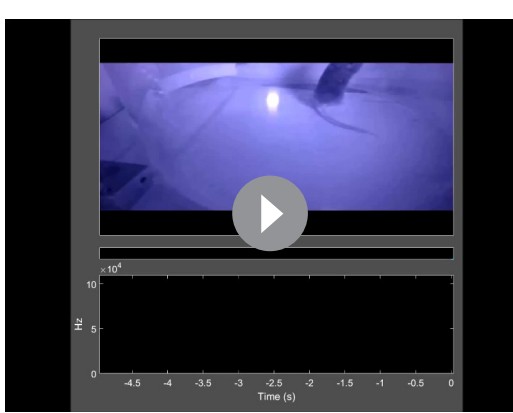

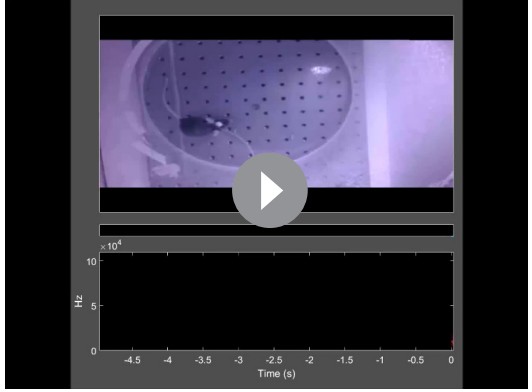

**Video 2.** Optogenetic activation of Amg$_{C/M-PAG}$ neurons causes no obvious behavioral effects in the absence of a social partner. An example male mouse with ChR2 expression in Amg$_{C/M-PAG}$ neurons is shown alone in a chamber with no social partner. Optogenetic activation of Amg$_{C/M-PAG}$ neurons with pulses of blue light does not elicit ultrasonic vocalization (USV) production or any other obvious behavioral response. Video is shown at the top, a spectrogram (bottom) showing the audio recording is synchronized to the video, and pitch-shifted audio (80 kHz to 5 kHz transformation) is included to place any USVs that may have occurred within the human hearing range.
https://elifesciences.org/articles/63493#video2

**Video 3.** Optogenetic activation of AmgC/M-PAG neurons causes no obvious behavioral effects in the absence of a social partner. An example male mouse with ChR2 expression in AmgC/M-PAG neurons is shown alone in a chamber with no social partner. Optogenetic activation of AmgC/M-PAG neurons with pulses of blue light does not elicit ultrasonic vocalization (USV) production or any other obvious behavioral response. Video is shown at the top, a spectrogram (bottom) showing the audio recording is synchronized to the video, and pitch-shifted audio (80 kHz to 5 kHz transformation) is included to place any USVs that may have occurred within the human hearing range.
https://elifesciences.org/articles/63493#video3

neurons in isolated mice. Optogenetic activation of Amg$_{C/M-PAG}$ neurons failed to elicit USV production and also did not drive any other obvious behavioral effects (*Videos 2*, *3*, *4*). However, when Amg$_{C/M-PAG}$ neurons were optogenetically activated in male mice that were actively courting females and vocalizing, USV production was immediately and reversibly suppressed (*Figure 4B*, N = 8 mice). This suppressive effect was restricted to the period when Amg$_{C/M-PAG}$ neurons were being optogenetically stimulated, and USV production rebounded following the end of the optogenetic stimulation period (*Figure 4B*). After using in situ hybridization to confirm that most Amg$_{C/M-PAG}$ neurons are GABAergic (~92% Amg$_{C/M-PAG}$ neurons express VGAT, *Figure 4C*), we used a similar intersectional viral strategy to express ChR2 selectively in GABAergic Amg$_{C/M-PAG}$ neurons (*Figure 4D*, AAV-retro-FLEX-ChR2 injected into the PAG of a VGAT-Cre mouse). With this strategy, we found that optogenetic activation of GABAergic Amg$_{C/M-PAG}$ neurons robustly suppressed the male's USV production during courtship encounters with a female (*Figure 4D*, N = 4 male mice). Finally, we tested the effects on vocal behavior of optogenetically activating the axon terminals of GABAergic Amg$_{C/M-PAG}$ neurons within the PAG (*Figure 4D*). Such bilateral terminal activation was also sufficient to suppress USV production (in N = 3 of 3 males; *Figure 4D*).

One possibility is that activating Amg$_{C/M-PAG}$ neurons suppresses USV production by putting the mouse into a fearful or aversive state, rather than through a direct suppressive effect of Amg$_{C/M-PAG}$ neurons on the PAG vocal gating circuit. To test this idea, we carefully examined the non-vocal behaviors of male mice during optogenetic activation of Amg$_{C/M-PAG}$ neurons. Mice exhibited neither freezing nor fleeing during optogenetic stimulation of Amg$_{C/M-PAG}$ neurons and, more notably, they usually continued to follow and sniff the female during the laser stimulation periods (*Video 5*; distance between male and female did not increase during optogenetic stimulation, *Figure 2—figure supplement 2*, panel B). We also confirmed that the change in USV production rates driven by the optogenetic activation of Amg$_{C/M-PAG}$ neurons was different from the change in spontaneous USV rates over time in mice that did not receive laser stimulation (*Figure 4D*, black trace), the change in USV rates over time in GFP control mice (*Figure 4D*, green trace), and the change in USV rates over time in Amg$_{C/M-PAG}$-ChR2-expressing mice that were connected to a dummy ferrule that only shined blue light over their head (*Figure 4D*, gray trace; p<0.01 for differences between ChR2 groups vs. control groups during laser time, p>0.05 for differences between groups in post-laser period; two-way ANOVA with repeated measures on one factor, p<0.01 for interaction between group and time, followed by post-hoc pairwise Tukey's HSD tests). Finally, we performed real-time place preference tests in which Amg$_{C/M-PAG}$ neurons were optogenetically activated when mice were in one of two sides of a test chamber (Amg$_{C/M-PAG}$ neurons were labeled with either the AAV-retro-Cre or the AAV-retro-ChR2 viral strategies). This experiment revealed that activation of Amg$_{C/M-PAG}$ neurons does not drive a negative place preference (*Figure 2—figure supplement 2*, panel A). In summary, activating Amg$_{C/M-PAG}$ neurons transiently and selectively suppresses USVs produced by male mice during courtship, an effect that cannot be accounted for by the mouse being put into a fearful or aversive state.

## Axonal projections of POA$_{PAG}$ and Amg$_{C/M-PAG}$ neurons

To further characterize the anatomy of POA$_{PAG}$ and Amg$_{C/M-PAG}$ neurons, we used intersectional methods to label these neurons with GFP and tdTomato respectively and traced their axonal

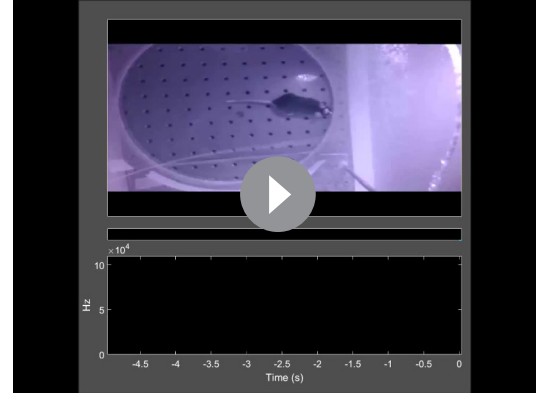

**Video 4.** Optogenetic activation of AmgC/M-PAG neurons causes no obvious behavioral effects in the absence of a social partner. An example male mouse with ChR2 expression in AmgC/M-PAG neurons is shown alone in a chamber with no social partner. Optogenetic activation of AmgC/M-PAG neurons with pulses of blue light does not elicit ultrasonic vocalization (USV) production or any other obvious behavioral response. Video is shown at the top, a spectrogram (bottom) showing the audio recording is synchronized to the video, and pitch-shifted audio (80 kHz to 5 kHz transformation) is included to place any USVs that may have occurred within the human hearing range.

https://elifesciences.org/articles/63493#video4

projections throughout the brain (*Figure 2—figure supplement 3*, AAV-retro-Cre injected into cau-dolateral PAG, AAV-FLEX-GFP into POA, AAV-FLEX-tdTomato into Amg$_{C/M}$). We observed dense projections from both POA$_{PAG}$ and Amg$_{C/M-PAG}$ neurons to a variety of dopaminergic cell groups, including the VTA, SNc and retrorubral/A8 region. We also note that Amg$_{C/M-PAG}$ neurons provide input to the lateral preoptic area (*Figure 2—figure supplement 3*, top left), while POA$_{PAG}$ neurons provide input to the same region in which Amg$_{C/M-PAG}$ cell bodies reside (*Figure 2—figure supplement 3*, middle left). As expected, we also observed dense and overlapping terminal fields from both of the cell groups within the caudolateral PAG (*Figure 2—figure supplement 3*, bottom right).

## Synaptic interactions between POA$_{PAG}$ and Amg$_{C/M-PAG}$ neurons and the PAG vocal gating circuit

The functional and anatomical experiments described above establish that two different populations of inhibitory forebrain neurons provide input to the PAG vocal gating circuit, one of which (the POA) promotes USV production in the absence of any social cues, while the other (the Amg$_{C/M}$) suppresses spontaneous USVs produced by male mice during courtship. To understand how two differ-ent GABAergic and presumably inhibitory inputs to the PAG can exert opposing effects on vocal behavior, we performed ChR2-assisted circuit mapping experiments in brain slices to characterize the properties of POA and Amg$_{C/M}$ synapses onto PAG-USV neurons and nearby GABAergic PAG neurons.

Given that optogenetic activation of GABAergic Amg$_{C/M-PAG}$ neurons suppresses USV produc-tion, we predicted that GABAergic Amg$_{C/M}$ neurons directly inhibit PAG-USV neurons. To test this idea, we performed whole-cell voltage clamp recordings from PAG-USV neurons while optogeneti-cally activating Amg$_{C/M-PAG}$ axons within the PAG. Briefly, AAV-FLEX-ChR2 was injected into the Amg$_{C/M}$ of a VGAT-Cre;Fos-dsTVA crossed mouse in order to express ChR2 in GABAergic Amg$_{C/M-PAG}$ axon terminals within the PAG. After four weeks, we used the CANE method (*Rodriguez et al., 2017*; *Sakurai et al., 2016*; *Tschida et al., 2019*) to infect PAG-USV neurons with a pseudotyped CANE-rabies virus driving the expression of mCherry (CANE-RV-mCherry, *Figure 5A–B*, see Materials and methods). We visually targeted our recordings to mCherry-expressing PAG-USV neu-rons and optogenetically activated Amg$_{C/M}$ terminals in the presence of TTX and 4AP in order to iso-late monosynaptic pathways (*Figure 5C–D*). Activating Amg$_{C/M-PAG}$ terminals evoked inhibitory postsynaptic currents (IPSCs) in a majority (16/29) of the mCherry-labeled PAG-USV neurons from which we recorded (mean current = 180.3 pA at 0 mV in TTX/4AP). These evoked IPSCs were completely abolished by application of the GABA$_A$ receptor antagonist gabazine (*Figure 5E–F*). No optogenetically elicited EPSCs were detected when recording at −70 mV, the chloride reversal potential. These findings support the idea that Amg$_{C/M-PAG}$ activity suppresses ongoing USV produc-tion by directly inhibiting PAG-USV neurons.

Given that activating GABAergic POA$_{PAG}$ neurons elicits vocalization (*Figure 2*), and that the majority of PAG-USV neurons are glutamatergic (*Tschida et al., 2019*), we hypothesized that POA$_{PAG}$ axons act via local GABAergic interneurons in the PAG to disinhibit PAG-USV neurons. To test this hypothesis, we first performed whole-cell patch clamp recordings from GABAergic PAG neurons while optogenetically activating POA$_{PAG}$ axons within the PAG. GABAergic PAG neurons were labeled by injecting AAV-FLEX-mCherry into the PAG of a VGAT-Cre mouse, while AAV-FLEXa-ChR2 was injected into the POA to express ChR2 in POA$_{PAG}$ axon terminals within the PAG (*Figure 6A–B*). After waiting 4 weeks to achieve functional expression of ChR2 in POA$_{PAG}$ axon ter-minals, we cut brain slices from these mice and recorded optogenetically evoked currents from fluo-rescently identified VGAT$^+$ PAG neurons (see Materials and methods). Optical stimulation of POA$_{PAG}$ axons with blue-light-evoked IPSCs in the majority (26/36) of voltage clamped GABAergic PAG neurons from which we recorded (mean current = 328.8 pA at 0 mV) (*Figure 6C*). These evoked IPSCs persisted upon application of TTX/4AP and were blocked by gabazine, indicating that POA$_{PAG}$ axons make inhibitory synapses directly onto GABAergic PAG neurons (*Figure 6D*).

To test whether these GABAergic PAG neurons synapse onto PAG-USV neurons, as predicted of a disinhibitory circuit mechanism, we injected AAV-FLEX-ChR2 into the PAG of a VGAT-Cre;TVA crossed mouse in order to express ChR2 in local VGAT$^+$ neurons (*Figure 6E*). After 2 weeks, we used CANE to selectively infect PAG-USV neurons with CANE-RV-mCherry (*Figure 6*). Several days later, we visually targeted mCherry-expressing PAG-USV neurons for whole-cell recordings while optogenetically activating local GABAergic PAG neurons in the presence of TTX and 4AP

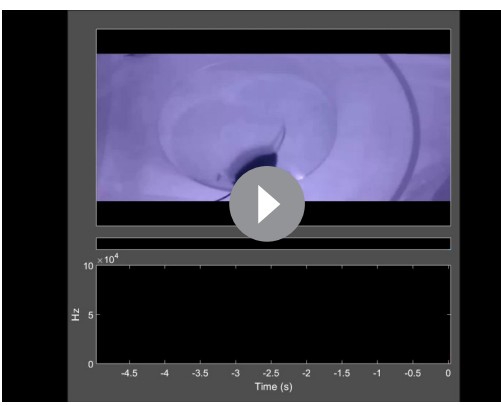

**Video 5.** Optogenetic activation of Amg$_{C/M\text{-PAG}}$ neurons transiently suppresses ultrasonic vocalization (USV) production. A male mouse which has ChR2 expressed in Amg$_{C/M\text{-PAG}}$ neurons is shown interacting with and producing USVs directed at a female social partner. Optogenetic activation of these neurons with pulses of blue light transiently suppresses USV production without suppressing non-vocal courtship behavior. Video is shown at the top, a spectrogram (bottom) showing the optogenetically elicited USVs is synchronized to the video, and pitch-shifted audio (80 kHz to 5 kHz transformation) is included to place the USVs within the human hearing range.
https://elifesciences.org/articles/63493#video5

(**Figure 6F**). Optogenetically activating local VGAT$^+$ neurons evoked IPSCs in almost all (13/16) of the PAG-USV neurons from which we recorded (mean current = 579.9 pA at 0 mV in TTX/4AP) and these currents were completely abolished by application of gabazine (**Figure 6G–H**). This experiment confirms the presence of a functional connection between local inhibitory neurons and the PAG-USV neurons that gate USV production.

We also performed whole-cell recordings from mCherry-labeled PAG-USV neurons while optogenetically activating POA$_{PAG}$ axons within the PAG (AAV-FLEX-ChR2 injected into the POA of a VGAT-Cre;Fos-dsTVA crossed mouse, CANE method used to infect PAG-USV neurons with CANE-RV-mCherry as described above; **Figure 6—figure supplement 1**, panel A, see Materials and methods). After first confirming that we could optogenetically evoke IPSCs in mCherry-negative cells in each slice, we visually targeted our recordings to mCherry-expressing PAG-USV neurons. Optogenetic activation of POA$_{PAG}$ terminals evoked IPSCs in only 1 of 23 PAG-USV neurons (**Figure 6—figure supplement 1**, panel B). Although there are caveats to interpreting a low probability of synaptic connection in brain slices, POA$_{PAG}$ neurons appear to provide fewer or weaker synaptic inputs to PAG-USV neurons than to nearby GABAergic PAG neurons,

supporting the idea that POA$_{PAG}$ neurons primarily act through PAG interneurons to disinhibit PAG-USV neurons and promote USV production.

## Discussion

Here, we used a combination of monosynaptic rabies tracing, optogenetic manipulations of neural activity in freely behaving animals, and optogenetics-assisted circuit mapping in brain slices to elucidate the functional relevance and synaptic organization of descending inputs to the PAG vocal gating circuit. We identified two populations of forebrain inhibitory neurons, one located in the preoptic hypothalamus and the other in a central-medial boundary zone within the amygdala, that drive opposing effects on vocal behavior. Optogenetic activation of POA$_{PAG}$ neurons drives robust and long-lasting bouts of vocalization in the absence of any social cues normally required to elicit vocalizations, and the acoustic features of optogenetically elicited USVs shared many features with spontaneously produced social USVs. In contrast, optogenetic activation of a VGAT$^+$ population of Amg$_{C/M\text{-PAG}}$ neurons transiently suppressed USV production in male mice during active courtship without disrupting other non-vocal courtship behaviors. Further, activation of Amg$_{C/M\text{-PAG}}$ neurons did not elicit fearful or aversive behavior, indicating that the effect on vocal behavior was not driven or accompanied by a generalized change in behavioral state. Finally, we paired optogenetic activation of descending POA or Amg$_{C/M}$ inputs to the PAG with whole-cell recordings from PAG-USV or GABAergic PAG neurons to investigate how these POA and Amg$_{C/M}$ inputs drive opposing effects on vocal behavior. These slice experiments support a model in which Amg$_{C/M\text{-PAG}}$ neurons directly inhibit PAG-USV neurons to suppress vocalization, while POA$_{PAG}$ neurons directly inhibit GABAergic PAG interneurons, which in turn inhibit PAG-USV neurons, resulting in a net disinhibition of PAG-USV neurons that promotes vocalization (**Figure 7**). To our knowledge, this is the first study to reveal the synaptic and circuit logic by which forebrain afferents to the PAG influence the decision to vocalize, a key behavior for communication and survival.

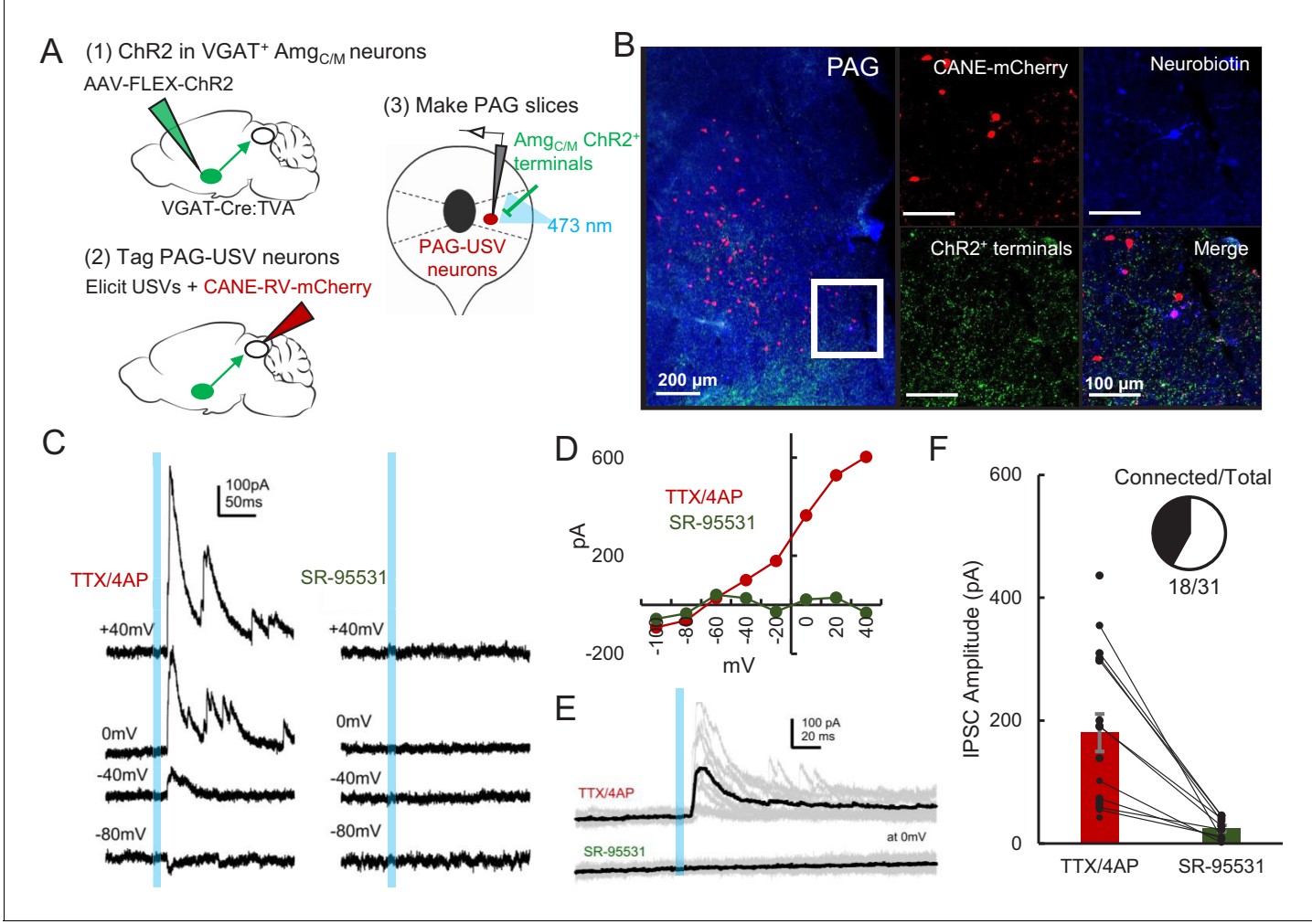

**Figure 5.** Amg$_{C/M}$ neurons provide direct inhibition onto PAG-USV neurons. (**A**) Viral strategy (left) and schematic (right) for whole-cell patch clamp recordings from fluorescently identified CANE-tagged PAG-USV neurons while optogenetically activating Amg$_{C/M-PAG}$ axons. (**B**) Example image of overlap of neurobiotin and mCherry-labeled PAG-USV cells with ChR2-expressing Amg$_{C/M-PAG}$ axon terminals in the PAG. (**C**) Example of light-evoked IPSCs at different voltages from one PAG-USV cell recorded in TTX/4AP while stimulating Amg$_{C/M-PAG}$ axons (left). Inhibitory postsynaptic currents (IPSCs) were abolished by bath gabazine application (right). (**D**) The peak magnitude of light-evoked currents at different membrane voltages for the same cell as (**C**) shows that the current reverses around the reversal potential of chloride and is abolished by gabazine. Currents were identified as IPSCs in this manner based on their reversal behavior and, for a subset of cells, by disappearance in gabazine. (**E,F**) Light-evoked IPSCs recorded in TTX/4AP (observed in n = 16 of 29 CANE-tagged cells from nine mice) were abolished by application of gabazine (n = 10 cells also recorded in gabazine, N = 10 cells, p<0.001, paired t-test). IPSC amplitude refers to the peak of the light-evoked current at 0 mV holding potential. Error bars represent S.E.M. See also *Figure 5—source data 1*.

The online version of this article includes the following source data for figure 5:

**Source data 1.** Source data for *Figure 5D and F*.

We observed that, when optogenetically activated, POA$_{PAG}$ neurons act directly through the PAG to elicit USV production in both male and female mice (*Figure 2*), confirming and extending a recent report that activation of GABAergic POA neurons elicits USVs in both sexes (*Gao et al., 2019*). These findings contrast with the behavioral observation that female mice in general produce fewer USVs than males. For example, female mice produce only about 1/5 of the total USVs recorded during male-female courtship interactions (*Neunuebel et al., 2015*), and we observed that female mice vocalize at lower rates than males when encountering novel female social partners (unpublished observations). Taken together, these findings suggest that different levels of POA$_{PAG}$ activity in males and females might contribute to sex differences in vocal behavior but, when strongly activated by optogenetic methods, POA$_{PAG}$ neurons in males and females are similarly potent in

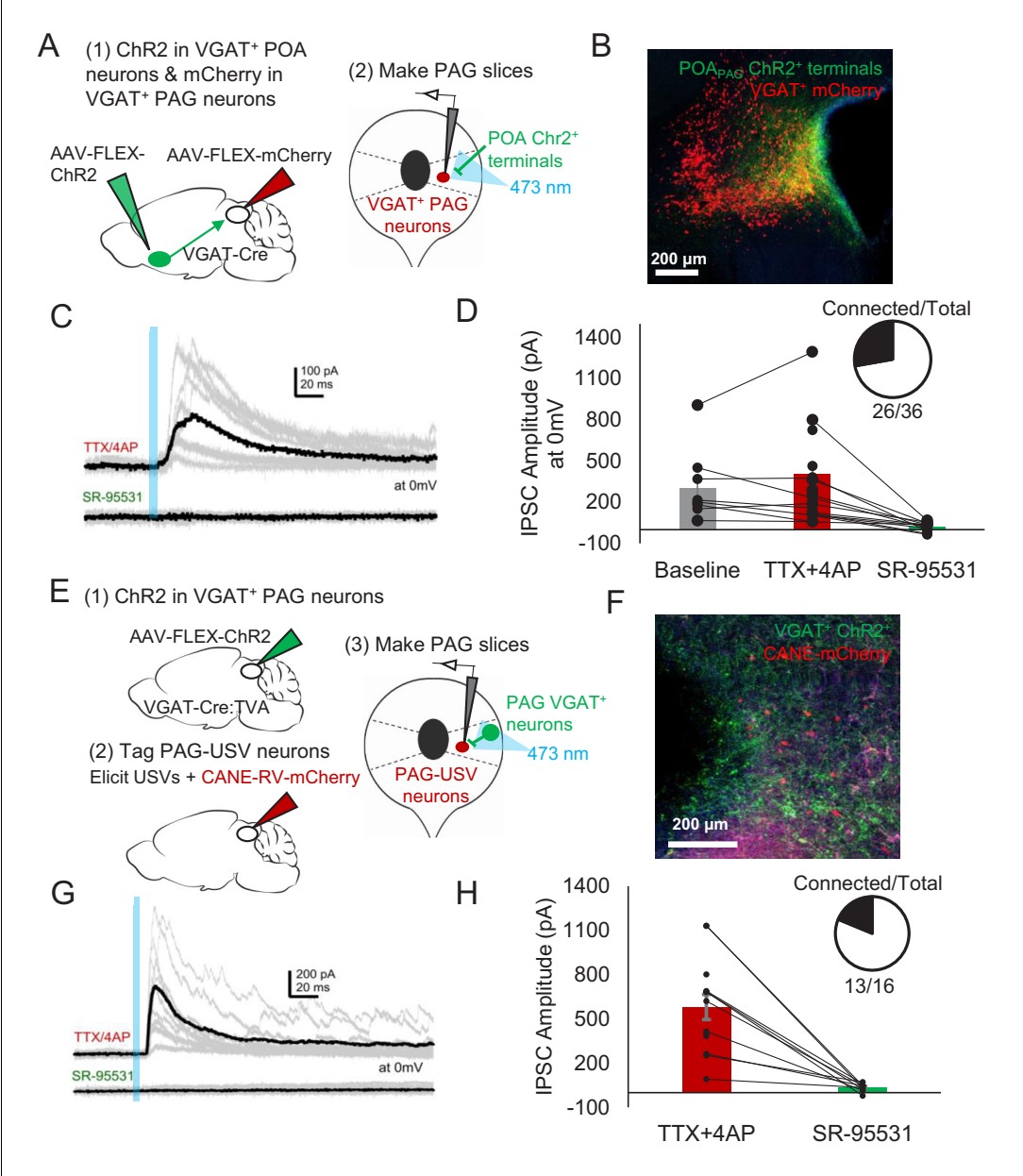

**Figure 6.** POA neurons provide direct inhibition onto VGAT+ PAG neurons, which provide direct inhibition onto PAG-USV neurons. (**A**) Viral strategy (left) and schematic (right) for whole-cell patch clamp recordings from fluorescently identified VGAT+ PAG cells while optogenetically activating POA_PAG axons. (**B**) Example image of mCherry-labeled VGAT+ neurons with ChR2-labeled POA_PAG axon terminals in the PAG. (**C, D**) Light-evoked inhibitory postsynaptic currents (IPSCs; observed in n = 26 of 36 VGAT+ neurons recorded from 11 mice) persisted in TTX/4AP and were abolished by bath application of gabazine (n = 10 cells recorded at baseline, n = 22 cells recorded in TTX/4AP, and n = 13 cells also recorded in gabazine including the following pairs: 6 cells recorded in both baseline and TTX/4AP, 3 cells recorded in both baseline and gabazine, and 10 cells recorded in both TTX/4AP and gabazine, p=0.03, one-way ANOVA comparing baseline vs. TTX+4-AP vs. SR-95531, followed by a post-hoc t-test revealing a significant difference between TTX+4-AP vs. SR-95531, p<0.018). IPSC amplitude refers to the peak of the light-evoked current at 0 mV holding potential. Error bars represent S.E.M. (**E**) Viral strategy (left) and schematic (right) for whole-cell recordings from fluorescently identified CANE-tagged PAG-USV neurons while optogenetically activating local VGAT+ PAG neurons. (**F**) Example image of mCherry-labeled CANE-tagged PAG-USV neurons and ChR2-labeled VGAT+ PAG neurons. (**G,H**) Light-evoked IPSCs recorded in TTX/4AP (observed in n = 13 of 16 CANE-tagged cells from four mice) were abolished by gabazine application (N = 10 cells also recorded in gabazine, p<0.001, paired t-test). IPSC amplitude refers to the peak of the light-evoked current at 0 mV holding potential. Error bars represent S.E.M. See also *Figure 6—figure supplement 1* and *Figure 6—source data 1*. The online version of this article includes the following source data and figure supplement(s) for figure 6:

**Source data 1.** Source data for *Figure 6D and H*.
**Figure supplement 1.** POA neurons provide direct inhibition onto few PAG-USV neurons.
*Figure 6 continued on next page*

*Figure 6 continued*

**Figure supplement 1—source data 1.** Source data for panel B of *Figure 6—figure supplement 1*.

their ability to activate the downstream PAG vocal gating circuit and elicit USVs. Although it remains unknown which factors might drive differential activation of male and female POA$_{PAG}$ neurons, it is possible that sex differences in the density (*Campi et al., 2013*; *Gorski et al., 1978*; *Orikasa and Sakuma, 2010*; *Panzica et al., 1996*), synaptic organization (*Raisman and Field, 1971*), and gene expression patterns (*Moffitt et al., 2018*; *Xu et al., 2012*), including those of sex hormone receptors (*Cao and Patisaul, 2011*) of POA neurons, might all contribute to this sexually dimorphic behavior (for a review, see *Lenz et al., 2012*). More broadly, our findings add to a growing body of literature indicating that male and female brains contain latent circuits for sex-typical behaviors that can be unmasked by artificial neural activation but that are gated in a sex-specific manner during natural behavior (*Clyne and Miesenböck, 2008*; *Gao et al., 2019*; *Rezával et al., 2016*; *Wei et al., 2018*).

We also found that similar to activation of POA$_{PAG}$ neurons, optogenetic activation of Esr1$^+$ POA neurons was sufficient to elicit USV production. Although a previous study reported that activation of Esr1$^+$ POA neurons promotes mounting (*Wei et al., 2018*), we failed to observe mounting when we optogenetically activated either POA$_{PAG}$ neurons or Esr1$^+$ POA neurons in male and female mice. Although the reasons for this discrepancy remain uncertain, one possibility is that our use of lower intensity optical stimulation can account for this difference (3–5 mW, 10–20 Hz vs. 10 mW, 40 Hz in *Wei et al., 2018*), and that the level of Esr1$^+$ POA neuronal activation required to elicit USV production is lower than the threshold to elicit mounting. An interesting possibility is that different projection-defined subsets of Esr1$^+$ POA neurons contribute to distinct aspects of courtship behavior, similar to what has been described for the contribution of projection-defined subsets of galanin-expressing POA neurons to distinct aspects of parental behavior (*Kohl et al., 2018*; *Wu et al., 2014*). Notably, though, a recent study found that ablation of VGAT$^+$ POA neurons did not affect the numbers of social USVs produced by male and female mice, although the acoustic features of male courtship USVs were altered following ablation of these neurons (*Gao et al., 2019*). In contrast, ablation or silencing of POA neurons greatly reduces non-vocal consummatory courtship behaviors including mounting and ejaculation (*Bean et al., 1981*; *Floody, 1989*; *Wei et al., 2018*). These findings are consistent with the idea that POA$_{PAG}$ neurons promote the production of USVs during later stages of courtship (*Gao et al., 2019*), which differ acoustically from USVs produced in earlier phases of courtship (*Hanson and Hurley, 2012*; *Keesom et al., 2017*; *Matsumoto and Okanoya, 2016*; *White et al., 1998*). These findings also suggest that other neuronal populations that lie upstream of the PAG vocal gating circuit, and that are potentially interconnected with the

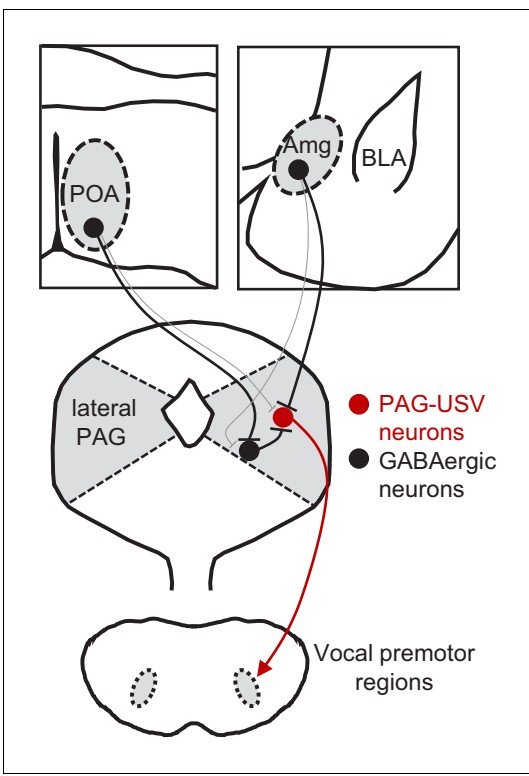

**Figure 7.** Model of bidirectional descending control of the periaqueductal gray (PAG) vocal gating circuit. Inhibitory neurons within the POA provide direct input to inhibitory neurons within the PAG, which in turn provide direct input to PAG-USV neurons. In this manner, activation of POA$_{PAG}$ neurons disinhibits PAG-USV neurons, which provide excitatory input to downstream vocal premotor neurons and drive USV production. Conversely, inhibitory neurons within the Amg$_{C/M}$ provide direct inhibitory input to PAG-USV neurons. Hence activation of Amg$_{C/M-PAG}$ neurons reduces PAG-USV activity and transiently suppresses USV production.

POA, serve to promote USV production during the early phases of courtship and in other behavioral contexts.

Here we employ a newly described VAE-based unsupervised modeling method to compare the acoustic features of optogenetically elicited USVs to each animal's repertoire of female-directed USVs. Although synchronous optogenetic activation of $POA_{PAG}$ neurons is likely quite different from the natural activity patterns of these neurons, the majority of optogenetically elicited vocalizations fall within the distribution of naturally produced USVs. This finding provides further experimental support for a model in which the PAG-USV neurons that are disinhibited by input from the POA gate USV production but do not directly pattern the acoustic content of vocalizations. The VAE also allowed us to identify and interrogate the acoustic features of optogenetically elicited vocalizations that fell outside the natural acoustic distribution. We found that these unusual optogenetically elicited USVs were louder and greater in frequency bandwidth, which we speculate may arise because synchronous optogenetic activation of POA neurons activates the PAG vocal gating circuit more strongly than occurs during natural behavior. However, whether altering the intensity or duration of POA stimulation systematically influences the acoustic features of optogenetically-elicited USVs remains to be tested. Another possibility is that because the mice in our experiments were singly tested in the absence of social partners and thus typically were not moving at high speeds during optogenetic stimulation, the USVs elicited by optogenetic stimulation of $POA_{PAG}$ neurons may be more similar acoustically to spontaneous USVs emitted in response to stationary social cues, such as female urine, rather than in response to a mobile female partner. Interestingly, a previous study found that the USVs produced by males in response to female urine were louder and had greater frequency bandwidth than those produced to female social partners (*Chabout et al., 2015*), reminiscent of the difference between female-directed and optogenetically evoked USVs in our dataset.

The current study also identifies a novel population of GABAergic $Amg_{C/M-PAG}$ neurons that lie at a boundary zone between the CeA and the medial amygdala and that project to PAG-USV neurons (i.e. $Amg_{C/M-PAG}$ neurons). Although this population of cells remains to be characterized comprehensively at a molecular and physiological level, our data show that transiently activating these neurons transiently suppresses USV production without driving fearful or aversive responses. Additionally, optogenetically activating $Amg_{C/M-PAG}$ neurons suppresses vocalization without interrupting non-vocal courtship behaviors more generally, providing additional support for the idea that PAG-USV cells are specialized neurons that gate USV production but that do not control non-vocal aspects of courtship.

We found that $Amg_{C/M-PAG}$ neurons make inhibitory synapses on PAG-USV neurons, which in turn gate vocalizations by exciting downstream vocal-respiratory pattern generating circuits (*Tschida et al., 2019*). Thus, the $Amg_{C/M}$ to PAG pathway provides a monosynaptic substrate through which vocalizations can be rapidly and effectively suppressed. We anticipate that such descending inhibitory inputs onto PAG-USV neurons act rapidly to suppress vocalization in behavioral contexts (in the presence of predators, conspecific competitors, etc.) in which vocalizing is risky or otherwise adverse, although this idea remains to be tested. We also note that while optogenetic activation of $Amg_{C/M-PAG}$ neurons transiently suppressed vocalization without obvious effects on non-vocal social behaviors and movement, it is possible that $Amg_{C/M}$ projections to other PAG cell types modulate diverse behaviors in addition to vocalization. Although $POA_{PAG}$ neurons are also GABAergic, we found that optogenetically activating these neurons promotes rather than suppresses USV production, likely through a disynaptic disinhibition of PAG-USV neurons mediated by local PAG interneurons. Consistent with the idea that disinhibition within the PAG is important for vocal production, work in primates has shown that pharmacological blockade of GABA receptors lowers the threshold for vocalization and elicits spontaneous vocalizations as well (*Forcelli et al., 2017*; *Jürgens, 1994*; *Lu and Jürgens, 1993*). Indeed, disinhibition of glutamatergic projection neurons has emerged as a prominent circuit motif within the PAG for releasing a variety of behaviors, including freezing (*Tovote et al., 2016*), pup grooming (*Kohl et al., 2018*), and antinociception (*Morgan and Clayton, 2005*). Our results support a model in which PAG-USV neuronal activity is tightly regulated by descending inputs as well as inputs from local GABAergic PAG neurons, which in turn integrate a variety of behaviorally relevant forebrain inputs to appropriately gate PAG-USV activity and hence USV production. More generally, such disinhibitory circuit motifs in the PAG may

provide a failsafe mechanism that carefully regulates the behavioral contexts in which crucial but potentially costly behaviors, including vocalization, are produced.

By exploiting selective genetic access to PAG-USV neurons as a point of entry into central circuits for social and courtship vocalizations, we have begun to map the brain-wide architecture and synaptic organization of circuitry for a complex, natural behavior. In addition to the inputs from the POA and Amg$_{C/M}$ that were the focus of this study, our transsynaptic tracing identified a number of forebrain regions whose projections converge onto the PAG vocal gating circuit, consistent with the idea that the PAG integrates a wide variety of social, environmental, and interoceptive information to gate vocalization in a context-appropriate manner. Given that context-dependent vocal gating is a hallmark of human vocalizations, including speech (*Stivers et al., 2009*), it will be of great interest in future studies to more fully describe the neuronal populations whose inputs to the PAG shape vocal behavior. We note that vocal behavior is not simply binary: in addition to deciding whether or not to vocalize, an animal must produce vocalizations that are appropriate for a given situation. The elucidation of circuit and synaptic mechanisms through which forebrain inputs to the PAG vocal gating circuit influence USV production represents an important first step toward understanding how forebrain-to-midbrain circuits regulate the production of vocalizations across different behavioral contexts to enable effective communication.

# Materials and methods

## Key resources table

| Reagent type (species) or resource | Designation | Source or reference | Identifiers | Additional information |
|---|---|---|---|---|
| Strain, strain background (*Mus musculus*, C57BL/6J) | C57 | Jackson Labs | RRID:IMSR_JAX:000664 | |
| Strain, strain background (*Mus musculus*, B6N.129S6(Cg)-*Esr1*$^{tm1.1(cre)And}$/J) | Esr1-Cre | Jackson Labs | RRID:IMSR_JAX:017911 | |
| Strain, strain background (*Mus musculus*, B6J.129S6(FVB)-*Slc32a1*$^{tm2(cre)Lowl}$/MwarJ) | VGAT-Cre | Jackson Labs | RRID:IMSR_JAX:016962 | |
| Strain, strain background (*Mus musculus*, B6;129S6-*Gt(ROSA)26Sor*$^{tim14(Cag-tdTomato)Hze}$/J) | Ai14 | Jackson Labs | RRID:IMSR_JAX:007908 | |
| Strain, strain background (*Mus musculus*, B6;129-*Fos*$^{tm1.1Fawa}$/J) | Fos-dsTVA | Jackson Labs | RRID:IMSR_JAX:027831 | |
| Recombinant DNA reagent | AAV2/1-hSyn-Flex-Chr2-eYFP | Addgene (K. Deisseroth) | RRID:Addgene_26973 | |
| Recombinant DNA reagent | AAV-pgk-retro-Cre | Addgene (P. Aebischer) | RRID:Addgene_24593 | |
| Recombinant DNA reagent | AAV2/1-pCAG-flex-GFP | Addgene (H. Zeng) | RRID:Addgene_51502 | |
| Recombinant DNA reagent | AAV2/1-pCAG-flex-Tdtomato | Addgene (H. Zeng) | RRID:Addgene_51503 | |
| Recombinant DNA reagent | AAV-flex-oG | Duke Viral Vector Core | | |

*Continued on next page*

*Continued*

| Reagent type (species) or resource | Designation | Source or reference | Identifiers | Additional information |
|---|---|---|---|---|
| Recombinant DNA reagent | EnvA-G-RV-GFP | *Rodriguez et al., 2017* (DOI: 10.1038/s41593-017-0012-1), *Sakurai et al., 2016* (DOI: 10.1016/j.neuron.2016.10.015) | | |
| Recombinant DNA reagent | CANE-RV-mCherry | *Rodriguez et al., 2017* (DOI: 10.1038/s41593-017-0012-1), *Sakurai et al., 2016* (DOI: 10.1016/j.neuron.2016.10.015) | | |
| Recombinant DNA reagent | AAV-flex-TVA-mCherry | *Rodriguez et al., 2017* (DOI: 10.1038/s41593-017-0012-1), *Sakurai et al., 2016* (DOI: 10.1016/j.neuron.2016.10.015) | | |
| Commercial assay or kit | HCR v3.0 | Molecular Instruments | | |
| Chemical compound, drug | Gabazine | Tocris | Cat# 1262 | (10 µM) |
| Chemical compound, drug | TTX | Tocris | Cat# 1069 | (2 µM) |
| Chemical compound, drug | 4AP | Sigma-Aldrich | Cat# 275875 | (100 µM) |
| Software, algorithm | MATLAB | Mathworks | RRID:SCR_001622 | |
| Software, algorithm | ImageJ | NIH | RRID:SCR_003070 | |
| Software, algorithm | ZEN | Zeiss | RRID:SCR_013672 | |
| Software, algorithm | Spike7 | CED | RRID:SCR_000903 | |
| Software, algorithm | pClamp | Molecular Devices | RRID:SCR_011323 | |
| Software, algorithm | IGOR Pro | WaveMetrics | RRID:SCR_000325 | |
| Other | NeuroTrace 435/455 | Invitrogen/Thermo Fischer Scientific | Cat# N21479 | (1:500) |

## Contact for reagent and resource sharing

Further information and requests for resources and reagents should be directed to the co-corresponding authors, Katherine Tschida (kat227@cornell.edu) or Richard Mooney (mooney@neuro.duke.edu).

## Experimental models and subject details

### Animal statement

All experiments were conducted according to a protocol approved by the Duke University Institutional Animal Care and Use Committee (protocol # A227-17-09).

## Animals

For optogenetic activation and axonal tracing experiments, the following mouse lines from Jackson labs were used: C57 (C57BL/6J, Jackson Labs, 000664), Esr1-Cre (B6N.129S6(Cg)-$Esr1^{tm1.1(cre)And}$/J, Jackson Labs, 017911), VGAT-Cre (B6J.129S6(FVB)-$Slc32a1^{tm2(cre)Lowl}$/MwarJ, Jackson Labs, 016962), Ai14 (B6;129S6-$Gt(ROSA)26Sor^{tim14(Cag-tdTomato)Hze}$/J, Jackson Labs, 007908). Fos-dsTVA mice (B6,129-$Fos^{tm1.1Fawa}$/J, Jackson Labs, 027831) were used for activity-dependent labeling of PAG-USV neurons employed in the transsynaptic tracing experiments and in whole-cell recording experiments. In a subset of whole-cell recording experiments, VGAT-Cre homozygous mice were crossed to Fos-dsTVA homozygous mice. Note that male Esr-1-Cre mice were often smaller and less healthy than their female littermates. While later weaning allowed them to grow to normal size, these animals still had lower survival rates after surgeries than any other animals used in this study, particularly when bilaterally implanting ferrules in the PAG for optogenetic stimulation of axon terminals.

## Method details

### Viruses

The following viruses and injection volumes were used: AAV2/1-hSyn-FLEX-ChR2-eYFP (Addgene), AAV-pgk-retro-Cre (Addgene), AAV-hsyn-retro-FLEX-ChR2 (Addgene), AAV-FLEX-GFP (Addgene), AAV-FLEX-tdTomato (Addgene), AAV-FLEX-oG (Duke Viral Vector Core). EnvA-ΔG-RV-GFP, CANE-RV-mCherry, and AAV-FLEX-TVA-mCherry were produced in house as previously described (*Rodriguez et al., 2017*; *Sakurai et al., 2016*; *Tschida et al., 2019*). The final injection coordinates were as follows: POA, AP = 0.14 mm, ML = 0.3 mm, DV = 5.5 mm; Amg$_{C/M}$, AP = −1.5 mm, ML = 2.3 mm, DV = 4.6 mm; PAG, AP = −4.7 mm, ML = 0.7 mm, DV = 1.75 mm. Viruses were pressure-injected with a Nanoject II (Drummond) at a rate of 4.6 nL every 15 s.

### Transsynaptic tracing from PAG-USV and GABAergic PAG neurons

To selectively infect PAG-USV neurons with viruses, ds-Fos-TVA males were given social experience with a female (30–60 min) that resulted in high levels of USV production (500–5000 USVs total). Males were then anesthetized (1.5–2% isoflurane), and the caudolateral PAG was targeted for viral injection. For transsynaptic tracing from PAG-USV neurons, the PAG was injected with a 4:1:1 mixture of CANE-LV-Cre, AAV-FLEX-TVA-mCherry, and AAV-FLEX-oG (total volume of 300 nL). After a wait time of 10–14 days, the PAG was then injected with EnvA-ΔG-RV-GFP (100 nL, diluted 1:5), and animals were sacrificed after waiting an additional 4–7 days.

To transsynaptically label inputs to GABAergic PAG neurons, the caudolateral PAG of VGAT-Cre mice was injected with a 1:1 mixture of AAV-FLEX-TVA-mCherry, and AAV-FLEX-oG (total volume of 100 nL). After a wait time of 10–14 days, the PAG was then injected with EnvA-ΔG-RV-GFP (100 nL, diluted 1:5), and animals were sacrificed after waiting an additional 4–7 days.

We note that because our goal was to identify long-range inputs onto PAG-USV and GABAergic PAG neurons, we used survival times that prioritized visualization of afferent cell bodies in distant locations rather than the integrity of the starter cell populations (which die off over time). Hence, we do not include quantification of starter cell populations within the PAG, as these cannot be meaningfully related to the numbers of cells that provide monosynaptic input to PAG-USV and GABAergic PAG neurons.

### In vivo optogenetic stimulation

Custom-made or commercially available (RWD) optogenetic ferrules were implanted in the same surgeries as viral injection just above target brain locations and were fixed to the skull using Metabond (Parkell). Neurons or their axon terminals were optogenetically activated with illumination from a 473 nm laser (3–15 mW) at 10–20 Hz (50 ms pulses, 2–10 s total) or with phasic laser pulses (1–2 s duration). Laser stimuli were driven by computer-controlled voltage pulses (Spike 7, CED). For stimulation of POA cell bodies or axon terminals, the laser was triggered manually at regular intervals while the animal was alone in the chamber. For stimulation of Amg$_{C/M}$ neurons or terminals, the laser was triggered manually each time the mouse began vocalizing for several seconds toward a female social partner.

## Post-hoc visualization of viral labeling

Mice were deeply anesthetized with isoflurane and then transcardially perfused with ice-cold 4% paraformaldehyde in 0.1 M phosphate buffer, pH 7.4 (4% PFA). Dissected brain samples were post-fixed overnight in 4% PFA at 4°C, cryoprotected in a 30% sucrose solution in PBS at 4°C for 48 hr, frozen in Tissue-Tek O.C.T. Compound (Sakura), and stored at –80°C until sectioning. To visualize viral labeling post-hoc, brains were cut into 80 μm coronal sections, rinsed 3x in PBS, and processed for 24 hr at four degrees with NeuroTrace (1:500 Invitrogen) in PBS containing 0.3% Triton-X. Tissue sections rinsed again 3 × 10 mins. in PBS, mounted on slides, and coverslipped with Fluoromount-G (Southern Biotech). After drying, slides were imaged with a 10x objective on a Zeiss 700 laser scanning confocal microscope.

## Floating section two-color in situ hybridization

In situ hybridization was performed using hybridization chain reaction (HCR v3.0, Molecular Instruments). Dissected brain samples were post-fixed overnight in 4% PFA at 4°C, cryoprotected in a 30% sucrose solution in RNAse-free PBS (i.e. DEPC-PBS) at 4°C for 48 hr, frozen in Tissue-Tek O.C.T. Compound (Sakura), and stored at –80°C until sectioning. 80 μm thick coronal floating sections were collected into a sterile 24-well plate in DEPC-PBS, fixed again briefly for 5 min in 4% PFA, then placed in 70% EtOH in DEPC-PBS overnight. Sections were rinsed in DEPC-PBS, incubated for 45 min in 5% SDS in DEPC-PBS, rinsed and incubated in 2x SSCT, pre-incubated in HCR hybridization buffer at 37°C, and then placed in HCR hybridization buffer containing RNA probes overnight at 37°C. The next day, sections were rinsed 4 × 15 min at 37°C in HCR probe wash buffer, rinsed with 2X SSCT, pre-incubated with HCR amplification buffer, then incubated in HCR amplification buffer containing HCR amplifiers at room temperature for ~48 hr. On the final day, sections were rinsed in 2x SSCT, counterstained with DAPI (Thermo Fisher, 1:5000), rinsed again with 2x SSCT, then mounted on slides and coverslipped with Fluoromount-G (Southern Biotech). After drying, slides were imaged with a 10x or 20x objective on a Zeiss 700 laser scanning confocal microscope.

Cells were scored from two to seven sections of tissue per brain region from each animal, and the absence or presence of staining within cells was quantified manually by comparing labeling within cells to background labeling in nearby regions known to have lower levels of expression of a given RNA transcript than the region of interest. Neighboring control regions with lower levels of transcript expression were present in the same coronal sections as the regions of interest and were determined by consulting the Allen Brain Atlas ISH Data (https://mouse.brain-map.org/search/index; experiment 72081554 for VGAT expression, experiment 79591677 for Esr1 expression, and experiment 73818754 for VGlut2 expression). These control regions were as follows for the following target regions and transcripts: (1a) POA VGAT: control region, fornix; (1b) Amg$_{C/M}$ VGAT: control region, thalamus; (1 c) CeA VGAT: control region, thalamus; (2a) POA Esr1: control region, fornix; (3a) POA VGlut2: control region, caudate putamen; (3b) Amg$_{C/M}$ VGlut2: caudate puteman; (3 c) CeA VGlut2: control region, caudate putamen.

## USV recording and analysis

To elicit USVs, single-housed males or females were presented with a freely moving female, either in a novel test chamber or in the home cage. USVs were recorded with an ultrasonic microphone (Avisoft, CMPA/CM16), amplified (Presonus TubePreV2), and digitized at 250 kHz (Spike 7, CED). USVs were detected using codes modified from the Holy lab (http://holylab.wustl.edu/) using the following parameters (mean frequency >45 kHz; spectral purity >0.3; spectral discontinuity <0.85; min. USV duration = 5 ms; minimum inter-syllable interval = 30 ms). To elicit USVs for tagging of PAG-USV neurons using CANE (for transsynaptic tracing and slice experiments), Fos$^{TVA}$ males were given social experience with a female (30–60 min session), either in their home cage fitted with an acoustically permeable lid or in a test chamber that had no lid and allowed easy microphone access. Sixty minutes from the start of the session, Fos$^{TVA}$ males were anesthetized and taken for injection of the PAG with viruses (see above), such that injections began approximately 2 hr from the start of USV production.

## Real-time place preference tests

Mice were lightly anesthetized to connect the 473 nm laser to the optogenetic ferrule, then mice were placed in the center of a custom-made two-sided test chamber, illuminated with infrared light only. The side of the chamber in which each mouse received optogenetic stimulation was chosen randomly for each place preference test. When the mouse was in the selected side, it received continuous 10 Hz optogenetic stimulation using the minimum laser power that had either elicited or inhibited USV production for that same mouse. Place preference was scored over a 20-min test period as the proportion of the total time that the mouse spent in the stimulated side of the chamber.

## Quantification of optogenetically elicited body movements

The mouse's position was measured using custom Matlab codes that detected and tracked the centroid of the mouse's body position across video frames (Logitech webcam, 30 frames per second), and speed of movement was calculated as the change in position across pairs of frames. To align movement with optogenetic activation of POA or $Amg_{C/M}$ neurons, we first estimated the temporal offset between the webcam video and USV audio by calculating the time of the peak cross-covariance between the high-pass filtered webcam audio and the low-pass filtered USV audio. This offset was then used to align the mouse's movement to the onset of each optogenetic laser stimulus. To measure the effects of optogenetic stimulation on the distance between an interacting male and female mouse, the position of each mouse was tracked manually in every $6^{th}$ frame, and the distance between mice was scored as the distance from the center of the male's head to the base of the female's tail.

## Comparison of acoustic features of optogenetically elicited USVs to female-directed USVs

A total of 52,821 USV syllables were segmented automatically with MUPET 2.0 using default parameter settings (*Van Segbroeck et al., 2017*). Of these syllables, 23,805 came from recordings of 15 mice recorded under both natural and optogenetic conditions (56% natural USVs). The remaining 29,016 syllables came from recordings of a control group of 10 mice used to establish across-day syllable repertoire variability. False positives (noise) from the experimental group were manually removed by visual inspection of spectrograms, with 79% of the original syllables retained. Syllables were analyzed using Autoencoded Vocal Analysis v0.2 (*Goffinet et al., 2019*), a Python package for generating low-dimensional latent descriptions of animal vocalizations using a VAE (*Kingma and Welling, 2013*). Briefly, the VAE jointly trains two probabilistic maps: an encoder and a decoder. Spectrograms are encoded into low-dimensional 'latent' representations which can be subsequently decoded to approximately reconstruct the original spectrograms. Both encoding and decoding distributions are parameterized by convolutional neural networks. We trained a VAE on spectrograms of single USV syllables from both experimental and control groups using the following parameters: min_freq = 30e3, max_freq = 110e3, nperseg = 1024, noverlap = 512, spec_min_val = −5.0, spec_max_val = −1.5, mel=False, time_stretch=True, within_syll_normalize=False. Each input spectrogram was 128-by-128 pixels (16,000 dimensions) and the VAE converged on a parsimonious representation of only five dimensions. To visualize these five-dimensional spaces, the latent representations of syllable spectrograms are projected into two dimensions using the UMAP algorithm (*McInnes et al., 2018*). To quantify differences in syllable repertoires, we estimate the Maximum Mean Discrepancy (*Gretton et al., 2012*) between distributions of latent syllable representations as in *Goffinet et al., 2019*. First, a baseline level of variability in syllable repertoire was established for each mouse by estimating MMD between the first and second halves of female-directed syllables emitted in a recording session. Then MMD between each mouse's natural and optogenetically elicited repertoires was estimated. A paired comparison test revealed significantly larger differences between optogenetic and natural repertoires than expected by variability within the natural condition recording sessions alone (two-sided, continuity-corrected Wilcoxon signed-rank test, W = 9, p<5e-3). We then estimated MMD between female-directed syllable repertoires recorded on different days, using the set of 10 control mice.

## Whole-cell recordings

Mice that received viral injections 2–4 weeks prior were deeply anesthetized with isoflurane and standard procedures were used to prepare 300-µm-thick coronal slices. The brain was dissected in ice-cold ACSF containing the following (in mM): 119 NaCl, 2.5 KCl, 1.30 $MgCl_2$, 2.5 $CaCl_2$, 26.2 $NaHCO_3$, 1.0 $NaHPO_4$-$H_2O$, and 11.0 dextrose and bubbled with 95% $O_2$/5% $CO_2$. The brain was mounted on an agar block and sliced in ice-cold ACSF with a vibrating-blade microtome (Leica). Slices were incubated for 15 min at 32°C in a bath of NMDG recovery solution containing the following (in mM): 93.0 NMDG, 2.5 KCl, 1.2 $NaH_2PO_4$, 30.0 $NaHCO_3$, 20.0 HEPES, 25.0 glucose, 2.0 thiourea, 5.0 Na L-ascorbate, 2.0 Na-pyruvate, 10.0 $MgSO_4$ $7H_2O$, 0.5 $CaCl_2$, and 95.0 HCl. Slices were then moved to a bath of HEPES storage solution containing the following (in mM): 93.0 NaCl, 2.5 KCl, 1.2 $NaH_2PO_4$, 30.0 $NaHCO_3$, 20.0 HEPES, 25.0 glucose, 2.0 thiourea, 5.0 Na L-ascorbate, 2.0 Na-pyruvate, 10.0 $MgSO_4$ $7H_2O$, and 0.5 $CaCl_2$, and allowed to gradually reach room temperature over the course of 1 hr, where they remained for the duration. Recordings were performed in ACSF at a temperature of 32°C. For voltage clamp experiments patch electrodes (4–8 MΩ) were filled with cesium internal solution containing the following (in mM): 130 cesium methanesulfonate, 5 QX-314 Br, 10 HEPES, 8 TEA-Cl, 0.2 EGTA, 4 ATP-Mg salt, 0.3 GTP-Na salt, and 10 phosphocreatine. Recordings were made using a Multiclamp 700B amplifier whose output was digitized at 10 kHz (Digidata 1440A). Series resistance was <25 MΩ and was compensated up to 90%. Signals were analyzed using Igor Pro (Wavemetrics). Neurons were targeted using interference contrast and epifluorescence to visualize fluorescent indicators previously expressed via viral injection. ChR2-expressing axon terminals were stimulated by 5–20 ms laser pulses (3–10 mW) from a 473 nm laser delivered via fiber optic inside the recording pipette (Optopatcher, A-M Systems). To confirm the direct nature of optogenetically evoked currents 2 µM TTX (Tocris) and 100 µM 4AP (Sigma-Aldrich) were added to the ACSF and perfused onto slices. To confirm that evoked currents were GABAergic, 10 µM gabazine (Tocris) was applied. Pharmacological agents including were bath applied for 10 min before making recordings.

## Code availability

All custom-written Matlab codes used in this study will be made publicly available at the Duke Digital Repository. The latest version of Autoencoded Vocal Analysis, the Python package used to generate, plot, and analyze latent features of mouse USVs, is freely available online: https://github.com/jackgoffinet/autoencoded-vocal-analysis.

## Quantification and statistical analyses

### Statistics

Parametric, two-sided statistical comparisons were used in all analyses unless otherwise noted (alpha = 0.05). No statistical methods were used to predetermine sample sizes. Error bars represent standard error of the mean unless otherwise noted. Mice were selected at random for inclusion into either experimental or control groups for optogenetic experiments. Mice were only excluded from analysis in cases in which viral injections were not targeted accurately, or in cases with absent or poor viral expression.

## Acknowledgements

Thanks to Michael Booze and Bao-Xia Han for additional mouse husbandry, thanks to Jun Takatoh for help with HCR in situ hybridization, and thanks to Shengli Zhao for providing CANE-related viruses. This work is supported by NIH grants DC 013826 (to RM) and MH 117778 (to FW and RM).

## Additional information

### Funding

| Funder | Grant reference number | Author |
| --- | --- | --- |
| National Institutes of Health | R01 DC013826 | Richard Mooney |
| National Institutes of Health | R01 MH117778 | Fan Wang |

| | | Richard Mooney |
| National Institutes of Health | F31 DC017879 | Valerie Michael |

The funders had no role in study design, data collection and interpretation, or the decision to submit the work for publication.

## Author contributions
Valerie Michael, Conceptualization, Data curation, Software, Formal analysis, Investigation, Visualization, Methodology, Writing - original draft, Writing - review and editing; Jack Goffinet, John Pearson, Software, Formal analysis, Methodology; Fan Wang, Conceptualization, Resources, Supervision, Funding acquisition, Project administration, Writing - review and editing; Katherine Tschida, Conceptualization, Data curation, Software, Formal analysis, Supervision, Investigation, Visualization, Methodology, Writing - original draft, Project administration, Writing - review and editing; Richard Mooney, Conceptualization, Resources, Supervision, Funding acquisition, Writing - original draft, Project administration, Writing - review and editing

## Author ORCIDs
Valerie Michael (iD) https://orcid.org/0000-0003-3288-9409
Jack Goffinet (iD) https://orcid.org/0000-0001-6729-0848
John Pearson (iD) https://orcid.org/0000-0002-9876-7837
Katherine Tschida (iD) https://orcid.org/0000-0002-8171-1722
Richard Mooney (iD) https://orcid.org/0000-0002-3308-1367

## Ethics
Animal experimentation: All experiments were conducted according to protocols approved by the Duke University Institutional Animal Care and Use Committee protocol (# A227-17-09).

## Decision letter and Author response
Decision letter https://doi.org/10.7554/eLife.63493.sa1
Author response https://doi.org/10.7554/eLife.63493.sa2

# Additional files

## Supplementary files
• Transparent reporting form

## Data availability
Data have been deposited to the Duke Research Data Repository, under the https://doi.org/10.7924/r4cz38d99. We have deposited 4 types of data in the repository: (1) confocal microscope images of in situ hybridization, (2) audio and video files from the mice used in this study, (3) slice electrophysiology data, and (4) custom Matlab codes used for data analysis. All other data analyzed in this study are included in the manuscript and supporting files.

The following dataset was generated:

| Author(s) | Year | Dataset title | Dataset URL | Database and Identifier |
|---|---|---|---|---|
| Michael V, Goffinet J, Pearson J, Wang F, Tschida K, Mooney R | 2020 | Data and scripts from: Circuit and synaptic organization of forebrain-to-midbrain pathways that promote and suppress vocalization | https://doi.org/10.7924/r4cz38d99 | Duke Research Data Repository, 10.7924/r4cz38d99 |

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
