## [Decision Letter]

**Acceptance summary:**

Activity dependent circuit tracing tools were used to identify inputs to the periaqueductal gray neurons that gate the production of ultrasonic vocalizations in mice. Optogenetic manipulations showed that two of these inputs have opposing effects on vocalization, while parallel in vitro experiments revealed circuit features underlying these opposing effects.

**Decision letter after peer review:**

[Editors’ note: the authors submitted for reconsideration following the decision after peer review. What follows is the decision letter after the first round of review.]

Thank you for submitting your work entitled "Circuit and synaptic organization of forebrain-to-midbrain pathways that promote and suppress vocalization" for consideration by *eLife*. Your article has been reviewed by a Senior Editor, a Reviewing Editor, and two reviewers. The reviewers have opted to remain anonymous.

Our decision has been reached after consultation between the reviewers. Based on these discussions and the individual reviews below, we regret to inform you that your work will not be considered further for publication in *eLife*.

The reviewers found the idea of investigating inputs to the PAG and their role in vocal production to be interesting, and appreciated the attempts at doing this in a cell-type specific manner. However, there were serious concerns raised about the evidence provided for the paper's claims. In particular, quantification of and verification of the specificity of the results for both the anatomical and the optogenetic experiments were found to be lacking. These are critically necessary for supporting the conclusions regarding the roles of EA and POA in driving vocalizations.

Reviewer #1:

The manuscript by Michael et al. describes how two different regions of the forebrain, the extended amygdala (EA) and the preoptic region (POA) of the hypothalamus, connect and control the production of ultrasonic vocalizations (USVs) via the periaqueductal gray (PAG). USVs are produced during specific interactions with conspecifics and are fundamental for communication. To understand how different internal and external variables influence the production of USVs requires the identification of the pathways carrying information about such variables. This study makes important discoveries into our understanding of the circuits controlling USV production by identifying how two independent pathways communicate to PAG neurons that ultimately control USV production. Interestingly, these two pathways seem to both provide inhibition to the PAG region that controls the production of USVs, but the net effect is the opposite: while the POA promotes the production of vocalizations, the EA turns it off.

In a recent study (Tschida et al., 2019) the same group has identified a circuit in the caudolateral PAG circuit whose activity is necessary and sufficient to produce USVs, as well its output to the brainstem. Using a similar strategy that relies on the genetic labelling of PAG neurons involved in USV production, the authors now investigated how two different input regions may control the production of vocalizations. Through optogenetic manipulations and circuit mapping, the authors provide convincing information about the logic of connectivity between the amygdala/hypothalamus and the PAG, opening up the opportunity to understand how different internal and external variables might control this fascinating behavior. The study is well performed, using state of the art genetic tools to address the problem. However, in my opinion, the authors do not present or quantify a series of results that would make the paper much stronger. Also, despite the fact that the authors use state of the art tools to label PAG neurons that are involved in USV production (and which they described in Tschida et al., 2019), the manipulations performed in the input regions to the PAG are not specific. Together with the lack of some analysis on the reliability of the optogenetic manipulations performed, I wonder if some of the results are indeed caused by the direct manipulation of the EA and POA.

1) The authors start by identifying the monosynaptic inputs onto the PAG-USV neurons (labeled in an activity dependent manner) and onto GABAergic local interneurons (PAG-VGAT) which are in the vicinity of the PAG-USV and which might inhibit them. They found several structures in the forebrain, including the POA and the EA and CeA. I know these tools are routinely used, but it is also known that these tools can be highly variable and dependent on the volume of injection, number of starter cells, etc. There is no quantification of the starter cells and transsynaptic labeled cells for the experiments in Figure 1A and B. Are the animals used males or females? Also, the PAG is a huge structure, as well as the POA and the EA/CeA. I am not familiar with the term caudolateral PAG, but from the coordinates given, this seems very similar to the vlPAG described in Tovote et al., 2016. Is this true?

Also, the authors only show zoom ins from the labeled structures, POA and EA/CeA. Showing the labeled regions in the context of the rest of the brain would make the figure and results easier to understand. Also, from what I gather, the EA is comprised of the BST and CeA. I am confused that the authors do not consider the CeA part of the EA. Providing more information about the exact location of the labeled structures would make the paper much easier to follow.

These issues might seem irrelevant, but the devil is in the details in these experiments, as we know that the PAG is involved in many different behaviors and its architecture is very complex. And the same holds true for the POA and for the EA.

2) On Figure 1C/D the authors show the results of an immunohistological assay to determine the nature of the input to the PAG-USV and PAG-VGAT neurons. There is very little information in the methods about how this was quantified. The authors show results for only 2 mice in each condition. The numbers presented are quantified by analyzing how many slices per animal?

3) The authors use an optogenetic strategy to target the POA neurons that project to the caudolateral PAG (Figure 2). First, this strategy is markedly different from the one used in Figure 1. Primarily, this strategy (Cre dependent viruses and a retrograde AAV carrying CRE) allows to label all neurons in the POA that project to the PAG, not only the PAG-USV and the PAG-VGAT. It would be important to understand how the neurons labeled like this relate to the ones in the previous experiment.

Then they use ChR2 to elicit activity in the POA neurons and record USVs, showing that this manipulation leads to vocalizations in 6 out of 8 males and 3 out of 4 females. First, maybe I am missing something, but I do not see a control (animal implanted with a fiber and light only or animal injected with a control virus and implanted with a fiber and exposed to light). Second, the only data presented is the number of animals per sex that responded. It would be great to see the histology of these animals: was the injection or fiber mis-placed in the animals that it didn't work? Also, what does it mean to have 6 out of 8 males vocalizing? Every time the light was on did you observe the same result? What was the variability across trials? There is no quantification of the results except for some parameters presented in the supplementary data.

In the same Figure 2C/D it is shown the effect of the POA manipulation using the Esr1+ Cre line. Again, no quantification is offered. How do the results of the two POA manipulations compare to each other? The latency to start USVs seems quite different in the two experiments. But again, because only examples are shown, we do not know, what is the variability across experiments.

Are neurons labeled with strategy A and D sending collaterals to other regions? I also find it surprising that the results of stimulating directly the Esr1-POA-PAG terminals in the PAG gave rise to other behavioral results that were not observed when the somas were directly activated at the level of the PAG. How can you explain this?

How can you explain such long latencies for the vocalizations to start? In some cases, it took more than 5 seconds for the USVs to be produced (minimum latency). Besides the minimum latency, one should have an idea of the spread of the latency across animals and within animals. Why is the number of animals presented in Figure 2—figure supplement 2A different from Figure 2—figure supplement 2B? I am not familiar with this, but do animals vocalize in isolation? What was the schedule of optogenetic stimulation performed? Are you sure that the USVs were caused by the stimulation? With such long latencies and not knowing the spread in the latency within animal after ChR2 activation, I feel a bit uncomfortable making the strong claim that the POA stim is indeed the one causing the USVs. It is really hard to understand exactly the results.

Note: I think the data from Figure 2—figure supplement 2 should come in the main text.

All the POA-PAG experiments are done in a non-specific population, this is, not only to PAG-USV projecting neurons, but to all caudolateral PAG. Also, these neurons might have collaterals to other brain regions. Activation of the terminals might backpropagate and if those cells send collateral to other regions, the effect could be mediated by that. Could you try, in the experiment where you activated the terminals, to block activity of the POA? This way the results obtained by activating the terminals could be only attributed to the projections to the caudolateral PAG (similar to what was performed in Wong et al., 2016, where they applied TTX and lidocaine in the LS to block activity, while stimulating the LS terminals).

4) I am not familiar with the analysis performed to compare the opto induced versus female directed USVs. However, I was wondering if the authors tried to examine how females responded to the opto-USVs. Also, how do these vocalizations compare to the PAG-USV stimulation described in their recent paper? They mention that the latencies are much longer, but nothing mentioned in relationship to the other USV features.

5) The authors then explored the role of the Amygdala-PAG projections. The strategy designed now involves the use of Cre dependent viruses in the amygdala and the injection of retroAAVs-Cre in the PAG. With this strategy the authors mention that only the EA is labeled and not the CeA. Therefore, their manipulations are only of the EA. Again, I am not an expert in amygdala, but in many papers it is referred that the CeA is part of the EA and therefore I am confused. I stress this point because I am having a difficult time understanding the relationship between the areas manipulated in this study and Tovote et al., 2016. In that case, a similar circuit organization is described, where CeA neurons disinhibit vlPAG neurons to produce freezing.

So, I wonder if the results obtained in here can be in some way interpreted in a similar manner to the results obtained in that paper: that what you are inducing is a fear state that would lead to the animal stopping USV production. I know the authors analysed other behaviors and have no indication that the animal is entering a fearful state during the EA-PAG neurons. However, this strategy does not label EA neurons that project specifically to the PAG-USV neurons, it labels all EA-VGAT neurons that project to the caudolateral PAG. In order to make sure the effect is due to the direct inactivation of PAG-USV neurons, could the authors perform the same mono-synaptic tracing done in Figure 1 and then label the EA-PAG projecting neurons with another strategy? For example, are all neurons EA-PAG neurons gabaergic? If yes, they could just inject a non-flexed retroAAV in the PAG and then perform the monosynaptic tracing starting from the PAG-USV. If those two experiments labeled the same population in the EA, then they could argue that the effect of the opto stim of EA-PAG neurons is indeed by inhibiting PAG-USV neurons (with the opto manipulation being done at the terminals, in the PAG, like in Figure 4D).

Reviewer #3:

During many behavioral contexts mice elicit ultrasonic vocalizations (USV) and it has been shown that these utterances are mainly generated by subcortical areas. Specifically, the periaqueductal grey (PAG) is gating vocal production in mammals. Recently, the same lab has published a paper about optogenetically activating/suppressing a subgroup of PAG neurons that had been previously shown to be active during USVs. In the previous article they describe how silencing PAG-USV neurons blocks USV production whereas activating PAG-USV neurons promotes USV production. In this study the authors aim to investigate the anatomical upstream sources that provide input to the PAG-USV neurons and ask whether the upstream inputs are functional i.e. sufficient for exerting vocal production behaviors.

In short, the anatomical tracing of PAG-USV inputs is interesting but not entirely surprising since PAG inputs have been traced before. The cell-type specific tracing adds a new point. The behavioral experiments provide mixed insights into the circuitry. On the one hand they stand in contrast with previous reports and on the other hand they replicate the same findings which have been described in previous reports. This makes the current study less novel and appealing. In addition, data analysis is insufficient and consequently, I have major concerns to promote this study for publication in *eLife*.

The tracing of the synaptic inputs onto PAG-USV neurons was achieved via activity-dependent labeling. In addition to labeling these neurons, the authors also labeled the upstream inputs onto GABAergic PAG neurons which provide inhibition onto PAG-USV neurons. Anatomical connections of the PAG had been mapped previously and the novelty here lies in the possibility to determine which cell-type (PAG-USV or GABAergic PAG neuron) other areas project to. The authors completely miss out on this and only provide Table 1 which contains insufficient information to appreciate new insights. I am suggesting to supply example images and quantitative data instead of “+” and “-“ and to discuss these results further.

Although neurons were labeled in a range of different brain areas, the authors decided to focus on the hypothalamus and the amygdala. This choice remains elusive is not well motivated. In addition, the authors put up a strawman by arguing that opposing behavioral effects can be hypothesized since the hypothalamus has been shown to be involved in sexual behaviors whereas the amygdala is involved in fear-related behaviors. However, the hypothalamus is also implicated in anxiety behaviors and the amygdala in positive emotional behaviors.

Subsection “Inhibitory neurons in the hypothalamus and amygdala provide input to the PAG vocal gating circuit”: “dense labeling” – This is not clear from the data. In Figure 1 POA labeling seems sparse in both PAG-USV and PAG-VGAT+? Provide quantitative assessment.

The authors claim that optogenetic stimulation of POA-PAG neurons elicits USVs. The provided evidence is not sufficient to underline this claim. Figure 2A and the movie are single occurrences that were aligned with the optogenetic pulse. How can the authors exclude that this did not happen by chance? Also, the onset of the vocalization differs in Figure 2 and the movie. If the vocalizations occur on a regular basis after optogenetic stimulation these data must be shown and quantified. How reliable was this effect? What is the latency? In addition, Gao et al., 2018 already described that optogenetic activation of GABAergic neurons in POA can evoke USVs in mice which make the data not novel.

To narrow down the molecular phenotype of the POA neurons the authors performed in situ hybridization and determined that POA-PAG neurons expressed the Estrogen α receptor. Wei et al., 2018 demonstrate that activation of POA neurons expressing estrogen α receptors results in sexually biased displays. Michael and colleagues hint towards their inability to replicate this result. In subsection “Activating PAG-projecting POA neurons elicits USVs in the absence of social cues” the number of animals is presented but the data itself are not shown and the statistical tests are unclear. One way to address the mismatch between the Wei et al. study and the results shown here is to record from neurons during optogenetic stimulation. How can the authors confirm that the optogenetic stimulation results in a functional activation of the targeted neurons?

The presented timeline of optogenetic stimulation and resulting vocalizations (subsection “Activating PAG-projecting POA neurons elicits USVs in the absence of social cues”) is difficult to understand. Even when counting the different synaptic stages and adding significant conduction velocity delays, the time course is biologically not plausible but simply too long. This observation is not discussed in detail and it remains unclear. How can it be excluded that vocalizations are being produced by chance and that the correlation of stimulation and vocalization occurs by chance?

The authors attempt to quantify the USVs during optogenetic stimulation and in control conditions in an unbiased way and used an unsupervised modeling approach. The data were visualized as UMAPs of latent features (Figure 3C). Based on differences in the visual appearing of these maps and the MMD the authors claim that a subset of USVs are similar and another subset is dissimilar during opto-stimulation versus female presence (subsection “Acoustic characterization of USVs elicited by activation of POA neurons”). This statement is confusing, and it remains unclear what is similar or dissimilar. To address this issue the authors investigated the acoustics in more detail. They argue that opto-USVS tended to be louder and covered a higher frequency bandwidth. This result, displayed in Figure 3F, is difficult to grasp. In Figure 3F it cannot be discriminated whether points are overlaid and therefore, the green data points are not visible in most part of the figure. Another concern is that USVs differ in different social contexts. The authors should perform the same analysis on USVs that were elicited when no female was present and without opto-stimulation to ensure that the observed change is not due to difference in the social context. Temporal organization and usage can be differentiated in multiple putative USV classes also arising from distinct articulatory patterns (Castellucci et al., 2018).

Subsequently, Michael et al., tested the effect of stimulating PAG-projecting amygdala neurons and found that the stimulation of either EA-PAG neurons or even more specifically, GABAergic EA-PAG neurons results in the suppression of vocalizations during ongoing behavior. While this result is intriguing (Figure 4B) the presentation in Figure 4D (right panel) is misleading. Why is the USV count in the pre-condition so much higher than during the post condition? What is the effect of opto-stimulation on neural activity?

The authors add a section about the upstream axonal projections of POA-PAG and EA-PAG neurons. These data do not add to the focus of this study and are not discussed further. What is the point the authors want to make with this?

To test connectivity in detail the authors performed slice recordings and measured synaptic inputs onto different cell types while stimulating others. While they find that some PAG-USV neurons receive inhibitory currents when EA neurons are stimulated, the authors do not perform the symmetrical experiment and stimulate POA-PAG neurons while recording PAG-USV neurons. Instead, VGAT+ neurons are being recorded and it is shown that a subset receives inhibitory current as well. In addition, VGAT+ neurons are being stimulated and PAG-USV were recorded to demonstrate that VGAT+ neuron provide inhibition to these PAG-USV neurons. Unfortunately, the data are less conclusive as described in the text. For example, in subsection “Synaptic interactions between POAPAG and EAPAG neurons and the PAG vocal gating circuit” 'majority (18 out of 27)': Is this statistically significant? What can one compare it to? Is the number of recorded neurons too low?

Results section and Discussion section: “data not shown” – In times of reproducibility of data and open access this statement is unacceptable. If the authors want to add this information to the text and speculate about them, they have to be shown.

[Editors’ note: further revisions were suggested prior to acceptance, as described below.]

Thank you for submitting your article "Circuit and synaptic organization of forebrain-to-midbrain pathways that promote and suppress vocalization" for consideration by *eLife*. Your article has been reviewed by Catherine Dulac as the Senior Editor, a Reviewing Editor, and three reviewers. The reviewers have opted to remain anonymous.

The reviewers have discussed the reviews with one another and the Reviewing Editor has drafted this decision to help you prepare a revised submission.

Summary:

Michael et al., use activity dependent circuit tracing tools to identify inputs to PAG-USV neurons, which they previously showed (Tschida et al., 2019) play a crucial role in gating vocal behavior. They use optogenetic manipulations to show that two of these inputs have opposing effects on male courtship vocalizations, then use slice electrophysiology to describe the circuit logic by which these opposing effects come about. In their rebuttal, the authors have strengthened these findings with the addition of several control experiments and analyses. Though forebrain regions such as the hypothalamus and amygdala have previously been implicated in vocal production, presumably through their descending connections to the PAG, these experiments greatly expand the field of knowledge encompassing these circuits and synaptic mechanisms.

Your reviewers concur that your revised manuscript is greatly improved with the more in-depth quantification of the optogenetic manipulations.

Essential revisions:

1) One of the main claims of the paper is that there is an opposing effect of the POA versus Amg gabaergic input: while the POA leads to USVs, the Amg leads to USV suppression. However, if I understand correctly the experiment depicted in Figure 1B, the G and the TVA should only be expressed in gabaergic neurons of the PAG and then the rabies injection should label all neurons that project to those, irrespective of being gabaergic or glutamatergic (because the rabies is not Cre dependent). Then, the subsequent results with immunohistochemistry show that neurons in the POA and Amg are all gabaergic (according to the staining shown in Figure 1C and D). Later on, it is assumed that the POA input is disinhibitory, which makes sense with this result and everything that follows after. However, the fact that PAG GABAergic neurons also receive inhibitory input from the Amg is not pursued at all in the rest of the paper or discussed. Instead, it is assumed that the Amg GABAergic neurons only project to principal PAG neurons (which is depicted in the cartoon of Figure 7 as well). This should be discussed since it assumes that the projections of the Amg are much more complex than just inhibiting USVs. The cartoon should also, therefore, be revised.

2) Figure 2E: In my opinion, this is an important claim of the paper, that the projections from the POA to PAG can induce USVs. I don't think that such a strong claim can be done with a single male (this experiment is necessary to show that the effect is due to the projections of POA neurons in the PAG).

3) The addition to the Materials and methods section is useful: "Cells were scored from 2-7 sections of tissue per brain region from each animal, and the absence or presence of staining within cells was quantified manually by comparing labeling within cells to background labeling in nearby regions known to be negative for a given RNA transcript." But what are those control regions and how was it confirmed that these are "negative" for the given RNA transcript? In the interest of data transparency and reproducibility, this information should be included.

4) The authors state (subsection “Activating PAG-projecting AmgC/M neurons transiently suppresses USV production”) that "Optogenetic activation of AmgC/M-PAG neurons failed to elicit USV production and also did not drive any other overt behavioral effects". This seems like important evidence for the author's claim that the function of AmgC/M-PAG is primarily to suppress vocal behavior, but the data aren't shown. In the interest of data transparency and reproducibility, these data should be included as a supplemental figure.

---

## [Author Response]

[Editors’ note: the authors resubmitted a revised version of the paper for consideration. What follows is the authors’ response to the first round of review.]

Reviewer #1:The manuscript by Michael et al. describes how two different regions of the forebrain, the extended amygdala (EA) and the preoptic region (POA) of the hypothalamus, connect and control the production of ultrasonic vocalizations (USVs) via the periaqueductal gray (PAG). USVs are produced during specific interactions with conspecifics and are fundamental for communication. To understand how different internal and external variables influence the production of USVs requires the identification of the pathways carrying information about such variables. This study makes important discoveries into our understanding of the circuits controlling USV production by identifying how two independent pathways communicate to PAG neurons that ultimately control USV production. Interestingly, these two pathways seem to both provide inhibition to the PAG region that controls the production of USVs, but the net effect is the opposite: while the POA promotes the production of vocalizations, the EA turns it off.In a recent study (Tschida et al., 2019) the same group has identified a circuit in the caudolateral PAG circuit whose activity is necessary and sufficient to produce USVs, as well its output to the brainstem. Using a similar strategy that relies on the genetic labelling of PAG neurons involved in USV production, the authors now investigated how two different input regions may control the production of vocalizations. Through optogenetic manipulations and circuit mapping, the authors provide convincing information about the logic of connectivity between the amygdala/hypothalamus and the PAG, opening up the opportunity to understand how different internal and external variables might control this fascinating behavior. The study is well performed, using state of the art genetic tools to address the problem. However, in my opinion, the authors do not present or quantify a series of results that would make the paper much stronger. Also, despite the fact that the authors use state of the art tools to label PAG neurons that are involved in USV production (and which they described in Tschida et al., 2019), the manipulations performed in the input regions to the PAG are not specific. Together with the lack of some analysis on the reliability of the optogenetic manipulations performed, I wonder if some of the results are indeed caused by the direct manipulation of the EA and POA.1) The authors start by identifying the monosynaptic inputs onto the PAG-USV neurons (labeled in an activity dependent manner) and onto GABAergic local interneurons (PAG-VGAT) which are in the vicinity of the PAG-USV and which might inhibit them. They found several structures in the forebrain, including the POA and the EA and CeA. I know these tools are routinely used, but it is also known that these tools can be highly variable and dependent on the volume of injection, number of starter cells, etc. There is no quantification of the starter cells and transsynaptic labeled cells for the experiments in Figure 1A and B. Are the animals used males or females?

We have included additional images to supplement the transsynaptic tracing results currently provided in Supplementary file 1 (Figure 1—figure supplement 1, Figure 1—figure supplement 2, Figure 1—figure supplement 3, Figure 1—figure supplement 4). However, because our goal was to identify long-range inputs onto PAG-USV neurons, we used long survival times that prioritized visualization of afferent cell bodies in distant locations rather than the integrity of the starter cell populations (which die off over time). Thus, while we can count the few remaining starter cells, such quantification will not accurately reflect their original numbers. Thus, we would prefer not to provide such quantification, as it cannot be meaningfully related to the absolute numbers of cells that provide monosynaptic input to PAG-USV neurons. We have added language to this effect to the Materials and methods. In addition, we make no claims about the relative densities of these labeled afferents, rather we used functional approaches to explore their relevance to vocalization.

The transsynaptic tracing from GABAergic PAG neurons was carried out from N=2 females and N=4 males, and the two mice used for in situ hybridization (Figure 1C-D) were both males. Due to the difficulty in eliciting USVs from female mice, the transsynaptic tracing from PAG-USV neurons (Figure 1A) was performed exclusively in male mice, and we have clarified these details in the manuscript (subsection “Inhibitory neurons in the hypothalamus and amygdala provide input to the PAG vocal gating circuit”).

Also, the PAG is a huge structure, as well as the POA and the EA/CeA. I am not familiar with the term caudolateral PAG, but from the coordinates given, this seems very similar to the vlPAG described in Tovote et al., 2016. Is this true?

No, the region of the PAG is not the same as that described by Tovote et al. 2016. The caudolateral PAG abuts the vlPAG but is largely dorsal to the vlPAG. We would like to emphasize to the Reviewer that the part of the PAG that contains PAG-USV neurons as characterized in Tschida et al., 2019 and as considered in the current study is a nearby but distinct part of the PAG to the vlPAG studied by Tovote et al., 2016 and others. We have clarified this distinction in subsection “Inhibitory neurons in the hypothalamus and amygdala provide input to the PAG vocal gating circuit” Results section.

Also, the authors only show zoom ins from the labeled structures, POA and EA/CeA. Showing the labeled regions in the context of the rest of the brain would make the figure and results easier to understand.

Lower magnification views of rabies-labeled neurons within the POA and the Amg_C/M_/CeA are now shown in Figure 1—figure supplement 1.

Also, from what I gather, the EA is comprised of the BST and CeA. I am confused that the authors do not consider the CeA part of the EA. Providing more information about the exact location of the labeled structures would make the paper much easier to follow.

The confusion raised by reviewer 1 here is a result of our previous choice of name for the region of the amygdala that we characterize in this study. The pocket of cells labeled in the amygdala with both transsynaptic tracing (Figure 1B) and our retro-Cre viral strategy (Figure 4A and Figure 4—figure supplement 1) occupies a boundary zone between the medial amygdala and the central amygdala that did not have any name in any brain atlases that we examined. In the revised manuscript, we call this region the “central-medial boundary zone” (Amg_C/M_) to avoid this confusion.

2) On Figure 1C/D the authors show the results of an immunohistological assay to determine the nature of the input to the PAG-USV and PAG-VGAT neurons. There is very little information in the methods about how this was quantified. The authors show results for only 2 mice in each condition. The numbers presented are quantified by analyzing how many slices per animal?

The data shown in Figure 1C-D are of representative in situ hybridization experiments (i.e., RNA staining), and the absence or presence of staining was quantified manually. We have added this information to the Materials and methods, as well as information regarding the number of sections from which the total number of scored neurons came. Quantification for each in situ hybridization came from N=2 animals per condition, which is standard in the field.

3) The authors use an optogenetic strategy to target the POA neurons that project to the caudolateral PAG (Figure 2). First, this strategy is markedly different from the one used in Figure 1. Primarily, this strategy (Cre dependent viruses and a retrograde AAV carrying CRE) allows to label all neurons in the POA that project to the PAG, not only the PAG-USV and the PAG-VGAT. It would be important to understand how the neurons labeled like this relate to the ones in the previous experiment.

We agree with the reviewer that the AAV-retro-Cre viral strategy is different than the monosynaptic transsynaptic tracing from defined PAG cells types as shown in Figure 1. Unfortunately, we are not aware of a non-rabies-based viral strategy that would allow us to label monosynaptic inputs to defined PAG cell types for long-term labeling and optogenetic manipulations of activity. Because the rabies-based viral strategy kills the cells after the first week following infection, we are not able to apply this strategy to manipulate neural activity in forebrain inputs to defined PAG cell types. Given this limitation, we used the AAV-retro-Cre viral strategy which, as the reviewer notes, will label cells within a given region that project to the caudolateral PAG (with some exceptions, as noted with the absence of labeling within the CeA using the AAV-retro-Cre strategy, as detailed in Figure 4, Figure 4—figure supplement 1, and Figure 4—figure supplement 2).

The AAV-retro-Cre viral strategy will not label an identical set of cells as those labeled by transsynaptic tracing, and given that there is no way to apply both viral strategies within the same animal, it is not possible to directly compare the populations labeled with the two strategies. In general, however, it is reasonable to expect that the retro-Cre viral strategy will label forebrain neurons that provide input to the PAG vocalization circuit, in addition to forebrain neurons that provide input to other nearby cells in the caudolateral PAG.

Despite these practical limitations, we applied in situ hybridization to verify that the forebrain neurons labeled with the AAV-retro-Cre strategy are of the same neurotransmitter-type as those labeled with rabies-based tracing. In addition to observing robust effects on behavior of optogenetic activation of these PAG-projecting neurons, we also provide evidence that GABAergic EA neurons directly inhibit PAG-USV neurons (Figure 5) and that GABAergic POA neurons directly inhibit VGAT+ PAG neurons (Figure 6). Together, we argue that these findings provide strong support for the conclusion that preoptic and amygdala neurons labeled by the AAV-retro-Cre strategy provide functional and behaviorally relevant inputs to the PAG vocalization circuit.

Then they use ChR2 to elicit activity in the POA neurons and record USVs, showing that this manipulation leads to vocalizations in 6 out of 8 males and 3 out of 4 females. First, maybe I am missing something, but I do not see a control (animal implanted with a fiber and light only or animal injected with a control virus and implanted with a fiber and exposed to light).

The reviewer is correct that we did not include GFP controls for the POA optogenetic activation experiments. We now include data from 5 POA GFP control mice, and we also include quantification to illustrate that the USV rates elicited by optogenetic activation of POA_PAG_ neurons are significantly greater than baseline vocalization rates in these same animals.

Second, the only data presented is the number of animals per sex that responded. It would be great to see the histology of these animals: was the injection or fiber mis-placed in the animals that it didn't work? Also, what does it mean to have 6 out of 8 males vocalizing? Every time the light was on did you observe the same result? What was the variability across trials? There is no quantification of the results except for some parameters presented in the supplementary data.

As stated in the Materials and methods, we excluded any animals with mistargeted injections or poor viral expression from further analysis.

In the same Figure 2C/D it is shown the effect of the POA manipulation using the Esr1+ Cre line. Again, no quantification is offered. How do the results of the two POA manipulations compare to each other? The latency to start USVs seems quite different in the two experiments. But again, because only examples are shown, we do not know, what is the variability across experiments.

There is variability in the efficacy of optogenetic activation in eliciting vocalizations across trials for single animals, between animals within a condition, and between conditions as well. We now include detailed quantification of the mean number of USVs elicited per stimulation, proportion of stim trials that elicited USVs, and mean latency from stim onset to first USV onset, in both experimental and control animals (Figure 2F). We want to highlight that some quantification of the latencies to USV onset and durations of USV bouts elicited by optogenetic activation are compared across mice and across conditions in Figure 2—figure supplement 1.

Are neurons labeled with strategy A and D sending collaterals to other regions? I also find it surprising that the results of stimulating directly the Esr1-POA-PAG terminals in the PAG gave rise to other behavioral results that were not observed when the somas were directly activated at the level of the PAG. How can you explain this?

Yes, these neurons send collaterals to other regions, as shown in in Figure 2—figure supplement 3. When we optogenetically activate Esr1+ POA cell bodies, the placement of the ferrule directly over the POA limits the spread of light to POA cell bodies, despite the fact that the AAV-FLEX-ChR2 expression sometimes spreads to adjacent overlying structures, such as the BNST. In the case of the axonal activation experiments, the ferrule placed in the PAG can activate the axon terminals of any PAG-projecting Esr1+ neurons within or nearby to the POA that are infected by the virus. As such, we suspect that the additional behavioral effects observed in the axon terminal activation experiments relate to the small amount of off-target viral labeling in Esr1+ neurons nearby the POA.

How can you explain such long latencies for the vocalizations to start? In some cases, it took more than 5 seconds for the USVs to be produced (minimum latency). Besides the minimum latency, one should have an idea of the spread of the latency across animals and within animals.

For the POA cell body activation experiments, the majority of mice (13/16) exhibited a minimum latency from laser stimulation to USV production of <500 ms (data shown in Figure 2—figure supplement 1B). The POA lies upstream of the PAG, which itself lies upstream of both the premotor and motor neurons important to vocalization. We found previously (Tschida et al., 2019) that average latency from optogenetic activation of PAG-USV neurons to USV onset was approximately 500 ms (minimum latency ~30 ms), and it is thus not surprising that optogenetic activation of the POA elicits USVs at longer latencies than those observed during PAG-USV activation. We also note that mean latencies on the order of seconds from optogenetic activation of the hypothalamus to observed effects of behavior have been reported previously (see Figure 2I and Figure 3H from Wei et al., 2018, showing mean latency of ~5s from POA activation to mounting and pup retrieval, respectively; similarly, see Figure 4I from Lin et al., 2011 showing mean latency of ~4s from VMHvl activation to attack). We have added language to the Results to directly compare the latencies from POA activation to USV production observed in the current study to latencies reported in other work from the onset of hypothalamic activation to subsequent effects on behavior. We have also included additional quantification regarding the variability in the latency from optogenetic stimulation to USV onsets within animals (Figure 2F).

Why is the number of animals presented in Figure 2—figure supplement 2A different from Figure 2—figure supplement 2B?

We used a variety of laser train stimuli to optogenetically elicit vocalizations from each animal, and in Figure 2—figure supplement 1CFigure , we omitted 5 animals from which we didn’t have a sufficient number of trials with the exact stimulation shown (2s-long laser trains). We have clarified the omission of the 5 animals in the figure legend.

I am not familiar with this, but do animals vocalize in isolation? What was the schedule of optogenetic stimulation performed? Are you sure that the USVs were caused by the stimulation? With such long latencies and not knowing the spread in the latency within animal after ChR2 activation, I feel a bit uncomfortable making the strong claim that the POA stim is indeed the one causing the USVs. It is really hard to understand exactly the results.

Mice vocalize rarely or not at all in isolation and we are highly confident that optogenetic stimulation of POA causes vocalizations in isolated mice. Specifically, we established in Tschida et al., 2019, that mice vocalize very little, if at all, in the absence of any social partners or social cues (see Figure 3A from that paper), and this finding is consistent with the broad consensus in the field that mice produce USVs in a variety of social contexts but very few or no USVs when monitored in social isolation. We have provided new quantification throughout Figure 2 (Figure 2B,D,E) to demonstrate that baseline vocal rates outside of periods of optogenetic stimulation are low, as well as control data showing that delivery of blue light to the brain in mice GFP expression in the POA does not elicit vocalization (Figure 2F).

Note: I think the data from Figure 2—figure supplement 2 should come in the main text.

We have left Figure 2—figure supplement1 as is but have included additional quantification within the main Figure 2.

All the POA-PAG experiments are done in a non-specific population, this is, not only to PAG-USV projecting neurons, but to all caudolateral PAG. Also, these neurons might have collaterals to other brain regions. Activation of the terminals might backpropagate and if those cells send collateral to other regions, the effect could be mediated by that. Could you try, in the experiment where you activated the terminals, to block activity of the POA? This way the results obtained by activating the terminals could be only attributed to the projections to the caudolateral PAG (similar to what was performed in Wong et al., 2016, where they applied TTX and lidocaine in the LS to block activity, while stimulating the LS terminals).

We agree that back-propagation is a caveat of any optogenetic terminal experiment. Although we previously considered the POA inactivation experiment suggested by the reviewer, we did not pursue that approach as lidocaine silencing of POA cell bodies is not guaranteed to block the back-propagation of activity from the PAG terminals to other parts of the axonal arborizations of these cells and therefore does not entirely control for this issue.

We now include data that directly address the reviewer’s concern, however. We observed that activating the axon collaterals of Esr1+ POA cells in another region of the brain, the VTA, fails to elicit vocalizations (Figure 2F). This negative result strengthens the case that there is something special about the projection from POA to PAG, rather than recruitment of non-PAG targets of POA-PAG axons.

4) I am not familiar with the analysis performed to compare the opto induced versus female directed USVs. However, I was wondering if the authors tried to examine how females responded to the opto-USVs. Also, how do these vocalizations compare to the PAG-USV stimulation described in their recent paper? They mention that the latencies are much longer, but nothing mentioned in relationship to the other USV features.

We did not characterize the responses of females to optogenetically elicited vocalizations, as this is tangential to the focus of our study. We also have not included a VAE analysis of the data from Tschida et al., 2019, as it would be difficult to draw systemic conclusions from a comparison to those data due to the limited size of the PAG-USV dataset (N=2 mice) in our previous study.

5) The authors then explored the role of the Amygdala-PAG projections. The strategy designed now involves the use of Cre dependent viruses in the amygdala and the injection of retroAAVs-Cre in the PAG. With this strategy the authors mention that only the EA is labeled and not the CeA. Therefore, their manipulations are only of the EA. Again, I am not an expert in amygdala, but in many papers it is referred that the CeA is part of the EA and therefore I am confused. I stress this point because I am having a difficult time understanding the relationship between the areas manipulated in this study and Tovote et al., 2016. In that case, a similar circuit organization is described, where CeA neurons disinhibit vlPAG neurons to produce freezing.

In the new manuscript, we call this region the “central-medial boundary zone” (Amg_C/M_) to avoid this confusion.

So, I wonder if the results obtained in here can be in some way interpreted in a similar manner to the results obtained in that paper: that what you are inducing is a fear state that would lead to the animal stopping USV production. I know the authors analysed other behaviors and have no indication that the animal is entering a fearful state during the EA-PAG neurons. However, this strategy does not label EA neurons that project specifically to the PAG-USV neurons, it labels all EA-VGAT neurons that project to the caudolateral PAG. In order to make sure the effect is due to the direct inactivation of PAG-USV neurons, could the authors perform the same mono-synaptic tracing done in Figure 1 and then label the EA-PAG projecting neurons with another strategy? For example, are all neurons EA-PAG neurons gabaergic? If yes, they could just inject a non-flexed retroAAV in the PAG and then perform the monosynaptic tracing starting from the PAG-USV. If those two experiments labeled the same population in the EA, then they could argue that the effect of the opto stim of EA-PAG neurons is indeed by inhibiting PAG-USV neurons (with the opto manipulation being done at the terminals, in the PAG, like in Figure 4D).

We disagree with the reviewer’s suggestion that activation of Amg_C/M_-_PAG_ neurons induces a fearful state in mice. In fact, we directly addressed this possibility extensively (subsection “Activating PAG-projecting AmgC/M neurons transiently suppresses USV production”) and demonstrated that Amg_C/M_ activation elicits neither fleeing nor freezing (Figure 2—figure supplement 2B) and that optogenetic activation of Amg_C/M_-_PAG_ neurons does not elicit real-time place aversion (Figure 2—figure supplement 2A).

As previously discussed, it is not possible to perform transsynaptic tracing from defined PAG cell types and AAV-retro-Cre labeling from the PAG in the same animal. Unfortunately, we are unaware of a non-rabies-based (and hence non-toxic) viral strategy that affords us the long-term access to forebrain neurons that project to defined PAG cell types that would be necessary to perform the experiment suggested by the reviewer. We believe that the robust effects on behavior observed in the AAV-retro-Cre optogenetic activation experiments provide strong support for the idea that PAG-projecting neurons within the Amg_C/M_ suppress USV production, and our slice data provide additional support to the idea that at least some of the PAG-projecting Amg_C/M_ neurons provide monosynaptic input to PAG-USV neurons, providing a plausible synaptic substrate for our observed behavioral effects.

Reviewer #3:During many behavioral contexts mice elicit ultrasonic vocalizations (USV) and it has been shown that these utterances are mainly generated by subcortical areas. Specifically, the periaqueductal grey (PAG) is gating vocal production in mammals. Recently, the same lab has published a paper about optogenetically activating/suppressing a subgroup of PAG neurons that had been previously shown to be active during USVs. In the previous article they describe how silencing PAG-USV neurons blocks USV production whereas activating PAG-USV neurons promotes USV production. In this study the authors aim to investigate the anatomical upstream sources that provide input to the PAG-USV neurons and ask whether the upstream inputs are functional i.e. sufficient for exerting vocal production behaviors.In short, the anatomical tracing of PAG-USV inputs is interesting but not entirely surprising since PAG inputs have been traced before. The cell-type specific tracing adds a new point. The behavioral experiments provide mixed insights into the circuitry. On the one hand they stand in contrast with previous reports and on the other hand they replicate the same findings which have been described in previous reports. This makes the current study less novel and appealing. In addition, data analysis is insufficient and consequently, I have major concerns to promote this study for publication in eLife.The tracing of the synaptic inputs onto PAG-USV neurons was achieved via activity-dependent labeling. In addition to labeling these neurons, the authors also labeled the upstream inputs onto GABAergic PAG neurons which provide inhibition onto PAG-USV neurons. Anatomical connections of the PAG had been mapped previously and the novelty here lies in the possibility to determine which cell-type (PAG-USV or GABAergic PAG neuron) other areas project to. The authors completely miss out on this and only provide Table 1 which contains insufficient information to appreciate new insights. I am suggesting to supply example images and quantitative data instead of “+” and “-“ and to discuss these results further.

We have included additional images of monosynaptic rabies tracing from the PAG vocal gating circuit in Figure 1—figure supplement 1, Figure 1—figure supplement 2, Figure 1—figure supplement 3, Figure 1—figure supplement 4. We found that the regions labeled by transsynaptic tracing provide input to both PAG-USV neurons and nearby VGAT+ PAG neurons (in both males and females).

Although neurons were labeled in a range of different brain areas, the authors decided to focus on the hypothalamus and the amygdala. This choice remains elusive is not well motivated. In addition, the authors put up a strawman by arguing that opposing behavioral effects can be hypothesized since the hypothalamus has been shown to be involved in sexual behaviors whereas the amygdala is involved in fear-related behaviors. However, the hypothalamus is also implicated in anxiety behaviors and the amygdala in positive emotional behaviors.

We have provided additional context and rationale for focusing on the hypothalamus and the amygdala (subsection “Inhibitory neurons in the hypothalamus and amygdala provide input to the PAG vocal gating circuit”). We agree with the reviewer that both the hypothalamus and the amygdala have been implicated in a variety of complex and distinct emotional states and behaviors, and there is no doubt that each of these brain regions contains sets of neurons that can influence social behavior in complex and often orthogonal ways.

Subsection “Inhibitory neurons in the hypothalamus and amygdala provide input to the PAG vocal gating circuit”: “dense labeling” – This is not clear from the data. In Figure 1 POA labeling seems sparse in both PAG-USV and PAG-VGAT+? Provide quantitative assessment.

We have included additional images (Figure 1—figure supplement 1, Figure 1—figure supplement 2, Figure 1—figure supplement 3, Figure 1—figure supplement 4Figures) that will allow readers to compare the density of labeling within the POA as compared to other forebrain inputs to the PAG that were labeled via transsynaptic tracing. We focused on the preoptic area in part because we found pronounced effects on vocal behavior in our initial experiments with this region, not because the preoptic area contained the highest density of transsynaptically-labeled neurons of all regions providing input to the PAG vocalization circuit. We also note that the efficacy of rabies-based transsynaptic tracing declines substantially as the distance between the upstream region and target region increases, and hence, the true density of a given distal input might be underestimated by considering transsynaptic tracing alone.

The authors claim that optogenetic stimulation of POA-PAG neurons elicits USVs. The provided evidence is not sufficient to underline this claim. Figure 2A and the movie are single occurrences that were aligned with the optogenetic pulse. How can the authors exclude that this did not happen by chance? Also, the onset of the vocalization differs in Figure 2 and the movie. If the vocalizations occur on a regular basis after optogenetic stimulation these data must be shown and quantified. How reliable was this effect? What is the latency? In addition, Gao et al., 2018 already described that optogenetic activation of GABAergic neurons in POA can evoke USVs in mice which make the data not novel.

We have provided additional quantification of the data we collected from these mice including the reliability of the optogenetic activation in eliciting USVs, variability in the latency of these effects within and across mice, as well as data from a variety of controls (Figure 2F). We established in Tschida et al., 2019 (Figure 3A) that mice vocalize very little if at all in the absence of a social partner, and we now show clearly in Figure that optogenetic activation of the POA elicits levels of USV production significantly greater than baseline.

We are aware that Gao et al., 2019 have shown that optogenetic activation of GABAergic POA neurons elicits USV production, and we are pleased that our results are in line with this previous observation. We have extended this observation by demonstrating that activation of Esr1+/GABAergic POA neurons that project to the PAG is sufficient to elicit USVs. Given that GABAergic POA neurons comprise distinct subsets of neurons with heterogeneous projection targets within the brain, we feel that this additional detail is both novel and important, and we emphasize how our work relates to previous findings in subsection “Activating PAG-projecting POA neurons elicits USVs in the absence of social cues”.

To narrow down the molecular phenotype of the POA neurons the authors performed in situ hybridization and determined that POA-PAG neurons expressed the Estrogen α receptor. Wei et al., 2018 demonstrate that activation of POA neurons expressing estrogen α receptors results in sexually biased displays. Michael and colleagues hint towards their inability to replicate this result. In subsection “Activating PAG-projecting POA neurons elicits USVs in the absence of social cues” the number of animals is presented but the data itself are not shown and the statistical tests are unclear. One way to address the mismatch between the Wei et al. study and the results shown here is to record from neurons during optogenetic stimulation. How can the authors confirm that the optogenetic stimulation results in a functional activation of the targeted neurons?

Using our optogenetic stimulation parameters, we were able to elicit robust USV production but never observed mounting of a conspecific following optogenetic activation of POA_PAG_ neurons. Given the robust effect on vocal behavior of optogenetic activation of POA neurons (of which we have provided additional quantification and controls in Figure 2 to convince the reviewers of these effects), we are certain that the optogenetic stimulation parameters applied in our study were sufficient to reliably activate POA neurons. As stated in the Discussion, we believe that our use of lower light intensity and stimulation frequencies could account for the discrepancy between our findings and those of Wei et al.

The presented timeline of optogenetic stimulation and resulting vocalizations (subsection “Activating PAG-projecting POA neurons elicits USVs in the absence of social cues”) is difficult to understand. Even when counting the different synaptic stages and adding significant conduction velocity delays, the time course is biologically not plausible but simply too long. This observation is not discussed in detail and it remains unclear. How can it be excluded that vocalizations are being produced by chance and that the correlation of stimulation and vocalization occurs by chance?

The absence of vocalizations in various control experiments that we now include (Figure 2F) rules out a chance relationship. In the revised manuscript, we have provided quantification showing that the USV production is elicited selectively during and following periods of optogenetic activation, and that baseline rates of vocalization outside of laser stimulation periods were low in these animals (Figure 2B, D, E). We note that mean latencies on the order of seconds from optogenetic activation of the hypothalamus to observed effects of behavior have been reported in other studies as well (see Figure 2I and Figure 3H from Wei et al., 2018, showing mean latency of ~5s from POA activation to mounting and pup retrieval, respectively; similarly, see Figure 4I from Lin et al., 2011 showing mean latency of ~4s from VMHvl activation to attack), adding additional support to the biological plausibility of our observed latencies. Even with direct optogenetic activation of PAG-USV neurons, we observed mean latencies upwards of hundreds of milliseconds (Tschida et al., 2019). We have added mentions of these previous findings in relation to the latencies we observed in the current study (subsection “Activating PAG-projecting POA neurons elicits USVs in the absence of social cues”).

The authors attempt to quantify the USVs during optogenetic stimulation and in control conditions in an unbiased way and used an unsupervised modeling approach. The data were visualized as UMAPs of latent features (Figure 3C). Based on differences in the visual appearing of these maps and the MMD the authors claim that a subset of USVs are similar and another subset is dissimilar during opto-stimulation versus female presence (subsection “Acoustic characterization of USVs elicited by activation of POA neurons”). This statement is confusing, and it remains unclear what is similar or dissimilar. To address this issue the authors investigated the acoustics in more detail. They argue that opto-USVS tended to be louder and covered a higher frequency bandwidth. This result, displayed in Figure 3F, is difficult to grasp. In Figure 3F it cannot be discriminated whether points are overlaid and therefore, the green data points are not visible in most part of the figure.

The data in Figure 3F are not overlaid in a manner that obscures the visibility of green points, and the figure accurately represents the finding that a subset of optogenetically-elicited USVs were louder and had greater frequency bandwidth than female-directed USVs produced by the same mice.

Another concern is that USVs differ in different social contexts. The authors should perform the same analysis on USVs that were elicited when no female was present and without opto-stimulation to ensure that the observed change is not due to difference in the social context. Temporal organization and usage can be differentiated in multiple putative USV classes also arising from distinct articulatory patterns (Castellucci et al., 2018).

Mice produce few or no USVs when tested in the absence of social cues or partners. However, males will often vocalize to the presentation of female urine. The behaviors exhibited by a mouse while investigating urine are relatively similar to those performed by solo-tested mice in our optogenetic activation experiments (no chasing or mounting due to the absence of a social partner, lots of sniffing, and slow, exploratory movements around the test chamber). Interestingly, a previous study found that USVs produced by males in response to female urine were significantly louder and tended to have greater bandwidth than those produced to a live female social partner (Chabout et al., 2015). We thank the reviewer for their insight, and we have added language to the manuscript (Discussion) to highlight the possibility that USVs elicited by optogenetic activation of the POA may be more similar to those produced in response to stationary female cues than to a moving female social partner.

Subsequently, Michael et al., tested the effect of stimulating PAG-projecting amygdala neurons and found that the stimulation of either EA-PAG neurons or even more specifically, GABAergic EA-PAG neurons results in the suppression of vocalizations during ongoing behavior. While this result is intriguing (Figure 4B) the presentation in Figure 4D (right panel) is misleading. Why is the USV count in the pre-condition so much higher than during the post condition? What is the effect of opto-stimulation on neural activity?

Mice typically produce USVs in bouts that last for several seconds. As one progresses further in time through a USV bout, there is an increasing probability that a bout will end. Thus, the decay in USV rates over time (pre vs. laser vs. post) in control animals simply reflects the natural statistics of USV production and is expected. We have clarified this idea in the legend for Figure 4.

The authors add a section about the upstream axonal projections of POA-PAG and EA-PAG neurons. These data do not add to the focus of this study and are not discussed further. What is the point the authors want to make with this?

We included these descriptive data simply for the sake of providing more information regarding the projection patterns of the cells characterized in this study, in case this information is of use to other scientists.

To test connectivity in detail the authors performed slice recordings and measured synaptic inputs onto different cell types while stimulating others. While they find that some PAG-USV neurons receive inhibitory currents when EA neurons are stimulated, the authors do not perform the symmetrical experiment and stimulate POA-PAG neurons while recording PAG-USV neurons. Instead, VGAT+ neurons are being recorded and it is shown that a subset receives inhibitory current as well. In addition, VGAT+ neurons are being stimulated and PAG-USV were recorded to demonstrate that VGAT+ neuron provide inhibition to these PAG-USV neurons. Unfortunately, the data are less conclusive as described in the text. For example, in subsection “Synaptic interactions between POAPAG and EAPAG neurons and the PAG vocal gating circuit” “majority (18 out of 27)”: Is this statistically significant? What can one compare it to? Is the number of recorded neurons too low?

We now include data in which we performed whole-cell recordings in brain slices from PAG-USV neurons while optogenetically activation PAG-projecting POA neurons. We only observed direct synaptic connections in 1/23 cells, and these findings are now reported in Figure 6—figure supplement 1.

Regarding the second point, we have collected additional data from VGAT+ PAG neurons while stimulating POA inputs, although we do not believe that the original number of recorded neurons that we reported is too low. We now report that 26/36 VGAT+ PAG neurons receive direct synaptic input from PAG-projecting POA neurons (subsection “Synaptic interactions between POAPAG and AmgC/M-PAG neurons and the PAG vocal gating circuit”). The result that GABAergic PAG neurons receive direct inhibitory inputs from the POA is quite clear. In our experience using optogenetic circuit mapping techniques in brain slices, finding that two thirds of cells are responsive to optogenetic stimulation is a good yield. We note that such a yield is typical or better than typical for optogenetic slice experiments in PAG. For example, see Figure 2I in the Tovote et al., 2016 Nature paper in which IPSCs were detected in 50% (12/24) of recorded PAG neurons upon optogenetic stimulation of local inhibitory neurons. Extended data Figure 2 from the same paper reports that IPSCs were detected in 3 example PAG neurons in response to terminal stimulation. Obviously, 26/36 responsive cells are a majority and there are many reasons why some cells recorded in brain slices fail to respond to optogenetic stimulation. First, it is unlikely that the POA would provide input to every VGAT+ neuron the PAG, as different PAG inhibitory neurons may be important for different PAG-mediated behaviors. Second, preparing a brain slice severs many connections, while also killing cells and terminals, thus further reducing the number of connections we can detect. Consequently, observing a direct inhibitory connection in 2/3 of recorded cells constitutes good evidence that this connection exists.

Results section and Discussion section: “data not shown” – In times of reproducibility of data and open access this statement is unacceptable. If the authors want to add this information to the text and speculate about them, they have to be shown.

We have removed these mentions of data that are not shown.

[Editors’ note: what follows is the authors’ response to the second round of review.]

Essential revisions:1) One of the main claims of the paper is that there is an opposing effect of the POA versus Amg gabaergic input: while the POA leads to USVs, the Amg leads to USV suppression. However, if I understand correctly the experiment depicted in Figure 1B, the G and the TVA should only be expressed in gabaergic neurons of the PAG and then the rabies injection should label all neurons that project to those, irrespective of being gabaergic or glutamatergic (because the rabies is not Cre dependent). Then, the subsequent results with immunohistochemistry show that neurons in the POA and Amg are all gabaergic (according to the staining shown in Figure 1C and D). Later on, it is assumed that the POA input is disinhibitory, which makes sense with this result and everything that follows after. However, the fact that PAG GABAergic neurons also receive inhibitory input from the Amg is not pursued at all in the rest of the paper or discussed. Instead, it is assumed that the Amg GABAergic neurons only project to principal PAG neurons (which is depicted in the cartoon of Figure 7 as well). This should be discussed since it assumes that the projections of the Amg are much more complex than just inhibiting USVs. The cartoon should also, therefore, be revised.

The reviewer is correct that our rabies tracing revealed that inhibitory Amg_C/M-PAG_ neurons provide input to PAG-USV neurons as well as to nearby GABAergic neurons. We have revised Figure 7 to include projections of Amg_C/M-PAG_ neurons to both of these PAG cell types. Our optogenetic and slice data support the idea that Amg_C/M-PAG_ neurons suppress USV production via their direct inhibitory inputs to PAG-USV neurons, but it is possible that these cells modulate diverse behaviors via their inputs to other cell types within the PAG, and we have added language to the Discussion to highlight this possibility.

2) Figure 2E: In my opinion, this is an important claim of the paper, that the projections from the POA to PAG can induce USVs. I don't think that such a strong claim can be done with a single male (this experiment is necessary to show that the effect is due to the projections of POA neurons in the PAG).

We have updated the subsection “Animals” to explain that we observed low survival rates for the Esr1-Cre males when we attempted to bilaterally implant the PAG with optogenetic ferrules. In our hands, Esr1-Cre males were smaller and less healthy than their female littermates and had to be weaned later than other animals in order to survive post-weaning. Even with these steps, we still had a low survival rate for Esr1-Cre males after this particularly invasive surgery. At a certain point, due to constraints on the number of animals we had available and the ethical concerns of repeating this experiment when the animals fared so poorly, we made the determination to stop attempting bilateral PAG implants in Esr1-Cre males and rather to focus on rounding out the dataset with female mice.

3) The addition to the Materials and methods section is useful: "Cells were scored from 2-7 sections of tissue per brain region from each animal, and the absence or presence of staining within cells was quantified manually by comparing labeling within cells to background labeling in nearby regions known to be negative for a given RNA transcript." But what are those control regions and how was it confirmed that these are "negative" for the given RNA transcript? In the interest of data transparency and reproducibility, this information should be included.

Neighboring control regions with low levels of transcript expression (relative to regions of interest) were present in the same coronal sections as the regions of interest and were determined by consulting the Allen Brain Atlas ISH Data (https://mouse.brain-map.org/search/index; experiment 72081554 for VGAT expression, experiment 79591677 for Esr1 expression, and experiment 73818754 for VGlut2 expression). These control regions were as follows for the following target regions and transcripts: (1a) POA VGAT: control region, fornix; (1b) Amg_C/M_ VGAT: control region, thalamus; (1c) CeA VGAT: control region, thalamus; (2a) POA Esr1: control region, fornix; (3a) POA VGlut2: control region, caudate putamen; (3b) Amg_C/M_ VGlut2: caudate putamen; (3c) CeA VGlut2: control region, caudate putamen. This information has been added to the Materials and methods.

We’ve also modified the wording describing transcript expression in same-section control regions, from being “negative” for transcript expression to having “low levels” of transcript expression, to acknowledge that an absence of strong labeling via ISH may indicate zero expression of a given transcript.

4) The authors state (subsection “Activating PAG-projecting AmgC/M neurons transiently suppresses USV production”) that "Optogenetic activation of AmgC/M-PAG neurons failed to elicit USV production and also did not drive any other overt behavioral effects". This seems like important evidence for the author's claim that the function of AmgC/M-PAG is primarily to suppress vocal behavior, but the data aren't shown. In the interest of data transparency and reproducibility, these data should be included as a supplemental figure.

To illustrate that optogenetic activation of Amg_C/M-PAG_ neurons did not elicit obvious behavioral changes, we have provided three additional supplementary videos (Video 3, Video 4, Video 5) depicting three different mice in which we optogenetically stimulated Amg_C/M-PAG_ neurons while the mice were alone in a chamber. These videos also include spectrograms and pitch-shifted playback of the audio recordings. It is clear from these videos that the mice do not vocalize upon stimulation and there are no obvious behavioral effects of the stimulation. We would also like to draw the reviewer’s attention to Figure 2—figure supplement 2, in which we demonstrate that optogenetic activation of Amg_C/M-PAG_ neurons does not affect movement speed.